# LOCAL COMPOSITE SADDLE POINT OPTIMIZATION

**Site Bai**
Department of Computer Science
Purdue University
bai123@purdue.edu

**Brian Bullins**
Department of Computer Science
Purdue University
bbullins@purdue.edu

## ABSTRACT

Distributed optimization (DO) approaches for saddle point problems (SPP) have recently gained in popularity due to the critical role they play in machine learning (ML). Existing works mostly target smooth unconstrained objectives in Euclidean space, whereas ML problems often involve constraints or non-smooth regularization, which results in a need for composite optimization. Moreover, although non-smooth regularization often serves to induce structure (e.g., sparsity), standard aggregation schemes in distributed optimization break this structure. Addressing these issues, we propose Federated Dual Extrapolation (FeDualEx), an extra-step primal-dual algorithm with local updates, which is the first of its kind to encompass both saddle point optimization and composite objectives under the distributed paradigm. Using a generalized notion of Bregman divergence, we analyze its convergence and communication complexity in the homogeneous setting. Furthermore, the empirical evaluation demonstrates the effectiveness of FeDualEx for inducing structure in these challenging settings.

## 1 INTRODUCTION

A notable fraction of machine learning (ML) problems belong to saddle point problems (SPP), including adversarial robustness (Madry et al., 2018; Chen & Hsieh, 2023), generative adversarial networks (GAN) (Goodfellow et al., 2014), matrix games (Abernethy et al., 2018), multi-agent reinforcement learning (Wai et al., 2018), among others. These applications call for effective distributed saddle point optimization as their scale evolves beyond centralized learning. In typical distributed optimization (DO) approaches, a central server coordinates collaborative learning among clients through rounds of communication. In each round, clients learn a synchronized global model locally without sharing their private data, then send the model to the server for aggregation, usually through averaging (McMahan et al., 2017; Stich, 2019), to produce a new global model. The cost of communication is known to dominate the optimization process (Konečný et al., 2016).

Although preliminary progress has been made in distributed saddle point optimization (Beznosikov et al., 2020; Hou et al., 2021), we would note that machine learning problems are commonly associated with task-specific constraints or non-smooth regularization, which results in a need for composite optimization (CO). Moreover, a common purpose for non-smooth regularization is to induce structure. Typical ones include $\ell_1$ norm for sparsity and nuclear norm for low-rankness, which show up in examples spanning from classical LASSO (Tibshirani, 1996), sparse regression (Hastie et al., 2015) to deep learning such as adversarial example generation (Moosavi-Dezfooli et al., 2016), sparse GAN (Zhou et al., 2020), convexified learning (Sahiner et al., 2022; Bai et al., 2024) and others.

Meanwhile, Yuan et al. (2021) identified the "curse of primal averaging" in standard aggregation schemes of DO, where the specific regularization-imposed structure on the client models may no longer hold after direct averaging on the server. For instance, each client may be able to obtain a sparse solution, yet averaging the solutions across clients yields a dense solution. To address this issue for convex optimization, they adopted the dual averaging technique (Nesterov, 2009), but this approach is not specifically designed for SPP. Even in the sequential deterministic setting, dual averaging or mirror descent (Nemirovskij & Yudin, 1983) achieve only a $\mathcal{O}(1/\sqrt{T})$ rate for SPP (Bubeck et al., 2015), whereas extra-step methods achieve a $\mathcal{O}(1/T)$ rate (Nemirovski, 2004; Nesterov, 2007). At the same time, existing distributed methods for SPP fail to cover these composite scenarios and address associated challenges, as summarized in Table 1.

| Task | Method | Composite & Constrained & Non-Euclidean | Convergence Rate | Convexity Assumption |
|------|--------|------|------|------|
| Min | FedAvg (Khaled et al., 2020) | ✗ | $\frac{\beta B}{RK} + \frac{\sigma B^{\frac{1}{2}}}{M^{\frac{1}{2}}R^{\frac{1}{2}}K^{\frac{1}{2}}} + \frac{\beta^{\frac{1}{3}}\sigma^{\frac{2}{3}}B^{\frac{2}{3}}}{K^{\frac{1}{3}}R^{\frac{2}{3}}}$ | convex |
| | FedDualAvg (Yuan et al., 2021) | ✓ | $\frac{\beta B}{RK} + \frac{\sigma B^{\frac{1}{2}}}{M^{\frac{1}{2}}R^{\frac{1}{2}}K^{\frac{1}{2}}} + \frac{\beta^{\frac{1}{3}}G^{\frac{2}{3}}B^{\frac{2}{3}}}{R^{\frac{2}{3}}}$ | convex |
| | **FeDualEx (Ours)** | ✓ | $\frac{\beta B}{RK} + \frac{\beta^{\frac{1}{2}}G^{\frac{1}{2}}B^{\frac{3}{4}}}{K^{\frac{1}{4}}R^{\frac{3}{4}}} + \frac{\sigma B^{\frac{1}{2}}}{M^{\frac{1}{2}}R^{\frac{1}{2}}K^{\frac{1}{2}}} + \frac{\beta^{\frac{1}{3}}G^{\frac{2}{3}}B^{\frac{2}{3}}}{R^{\frac{2}{3}}}$ | convex |
| Min-Max | Extra Step Local SGD (Beznosikov et al., 2020) | ✗ | $B\exp\{-\frac{\alpha KR}{\beta}\} + \frac{\sigma^2}{MKR} + \frac{\beta^2\sigma^2}{\alpha^4 K^2 R^2}$ | $\alpha$-strongly convex-concave |
| | SCCAFFOLD-S (Hou et al., 2021) | ✗ | $\frac{\beta^2}{\alpha^2}B\exp\{-\frac{\alpha KR}{\beta}\} + \frac{\sigma^2}{MKR} + \frac{\beta^2\sigma^2}{\alpha^4 KR^2}$ | $\alpha$-strongly convex-concave |
| | **FeDualEx (Ours)** | ✓ | $\frac{\beta B}{RK} + \frac{\beta^{\frac{1}{2}}G^{\frac{1}{2}}B^{\frac{3}{4}}}{K^{\frac{1}{4}}R^{\frac{3}{4}}} + \frac{\sigma B^{\frac{1}{2}}}{M^{\frac{1}{2}}R^{\frac{1}{2}}K^{1}} + \frac{\beta^{\frac{1}{2}}G^{\frac{1}{2}}B^{\frac{3}{4}}}{R^{\frac{1}{2}}}$ | convex-concave |

Table 1: We list existing convergence rates on composite convex optimization and smooth saddle point optimization in distributed settings similar to ours. Notations are $R$: communication rounds; $K$: local steps; $\beta$: smoothness; $B$: diameter; $G$: gradient bound; $M$: clients; $\sigma^2$: gradient variance. FedAvg is also included as a reference. We further note that none of the work other than ours covers composite SPP. They are included only for completeness.

We present the distributed paradigm for composite saddle point optimization defined in (1). In particular, we propose Federated Dual Extrapolation (FeDualEx) (Algorithm 1), which builds on Nesterov's dual extrapolation (Nesterov, 2007), a classic extra-step algorithm suited for SPP. It carries out a two-step evaluation of a proximal operator (Censor & Zenios, 1992) defined by the Bregman Divergence (Bregman, 1967), which allows for SPP beyond the Euclidean space. To adapt to composite regularization, FeDualEx also draws inspiration from recent progress in composite convex optimization (Yuan et al., 2021) and adopts the notion of generalized Bregman divergence (Flammarion & Bach, 2017) instead, which merges the regularization into its distance-generating function. With some novel technical accommodations, we provide the convergence rate for FeDualEx under the homogeneous setting, which is, to the best of our knowledge, the first convergence rate for composite saddle point optimization under the DO paradigm. In support of the proposed method, we conduct numerical evaluations to verify the effectiveness of FeDualEx on composite SPP.

To further demonstrate the quality of the induced structure, we include the primal twin of FeDualEx based on mirror prox (Nemirovski, 2004), namely "Federated Mirror Prox (FedMiP)", as a baseline for comparison in Appendix H. This is in line with the dichotomy between Federated Mirror Descent (FedMiD) and Federated Dual Averaging (FedDualAvg) (Yuan et al., 2021), from which Yuan et al. (2021) identified the "curse of primal averaging" in DO, i.e., the specific regularization-imposed structure on the client models may no longer hold after primal averaging on the server. It highlights that FeDualEx naturally inherits the merit of dual aggregation from FedDualAvg. In addition, we analyze FeDualEx for federated composite convex optimization and show that FeDualEx recovers the same convergence rate as FedDualAvg under the convex setting.

Last but not least, by reducing the number of clients to one, we show for the sequential version of FeDualEx that the analysis naturally yields a convergence rate for stochastic composite saddle point optimization which, to our knowledge, is the first such algorithm for non-Euclidean settings and matches the $\mathcal{O}(\frac{1}{\sqrt{T}})$ rate in general stochastic saddle point optimization (Mishchenko et al., 2020; Juditsky et al., 2011). Further removing the noise from gradient estimates, FeDualEx still generalizes dual extrapolation to deterministic composite saddle point optimization with a $\mathcal{O}(\frac{1}{T})$ convergence rate that matches the smooth case and also the pioneering composite mirror prox (CoMP) (He et al., 2015) as presented in Table 2.

**Our Contributions:**

- We propose FeDualEx for distributed learning of SPP with composite possibly non-smooth regularization (Section 4.1). In support of the proposed algorithm, we provide a convergence rate for FeDualEx under the homogeneous setting (Section 4.2). To the best of our knowledge, FeDualEx is the first of its kind that encompasses composite possibly non-smooth regularization for SPP under a distributed paradigm, as shown in Table 1.

- Additionally, we showcase the structure-preserving (e.g., sparsity) advantage of FeDualEx achieved through dual-space averaging. In particular, we present its primal twin FedMiP as a baseline to highlight this contrast (Appendix H).

| Noise | Rate | Composite SPP | Smooth SPP |
|---|---|---|---|
| Deterministic | $\mathcal{O}\left(\frac{1}{T}\right)$ | CoMP (He et al., 2015) **Deterministic FeDualEx (Ours)** | Mirror Prox (Nemirovski, 2004) Dual Extrapolation (Nesterov, 2007) Accelerated Proximal Gradient (Tseng, 2008) |
| Stochastic | $\mathcal{O}\left(\frac{1}{\sqrt{T}}\right)$ | Extragradeint (Euclidean) (Mishchenko et al., 2020) **Sequential FeDualEx (Ours)** | Mirror Prox (Juditsky et al., 2011) **Sequential FeDualEx (Ours)** |

Table 2: Convergence rates for convex-concave SPP. The deterministic version of FeDualEx generalizes dual extrapolation (DE) to composite SPP, and the sequential version of FeDualEx generalizes DE to both smooth and composite stochastic saddle point optimization.

- FeDualEx produces several byproducts in the CO realm, as demonstrated in Table 2 : (1) The sequential version of FeDualEx leads to the stochastic dual extrapolation for CO and yields, to our knowledge, the first convergence rate for the stochastic optimization of composite SPP in non-Euclidean settings . (2) Further removing the noise leads to its deterministic version, with rates matching existing ones in smooth and composite saddle point optimization (Section 5).

- We demonstrate experimentally the effectiveness of FeDualEx on various composite saddle point tasks, including bilinear problems on synthetic data with $\ell_1$ and nuclear norm regularization, as well as the universal adversarial training of logistic regression with MNIST and CIFAR-10 (Section 6).

## 2 RELATED WORK

We provide a brief overview of some related work and defer extended discussions to Appendix B.

The distributed optimization paradigm we consider aligns with that in Local SGD (Stich, 2019), which is also the homogeneous setting of Federated Averaging (FedAvg) (McMahan et al., 2017). Stich (2019) provides the first convergence rate for FedAvg, and it has been improved with tighter analysis and also analyzed under heterogeneity (e.g., (Khaled et al., 2020; Woodworth et al., 2020b)). Recently, Yuan et al. (2021) extended FedAvg to composite convex optimization and proposed FedDualAvg that aggregates learned parameters in the dual space and overcomes the "curse of primal averaging" in federated composite optimization.

For SPP, Beznosikov et al. (2020) investigate the distributed extra-gradient method for strongly-convex strongly-concave SPP in the Euclidean space. Hou et al. (2021) propose FedAvg-S and SCAFFOLD-S based on FedAvg (McMahan et al., 2017) and SCAFFOLD (Karimireddy et al., 2020) for SPP, which yields similar convergence rate to (Beznosikov et al., 2020). In addition, Ramezani-Kebrya et al. (2023) study the problem from the information compression perspective with the measure of communication bits. Yet, the aforementioned works are limited to smooth and unconstrained SPP in the Euclidean space. The more general setting of composite SPP is only found in sequential optimization literature, where the representative composite mirror prox (CoMP) (He et al., 2015) generalizes the classic mirror prox (Nemirovski, 2004) yet keeps the $\mathcal{O}(\frac{1}{T})$ convergence rate. In the stochastic setting, Mishchenko et al. (2020) analyzed a variant of stochastic mirror prox (Juditsky et al., 2011), which is then capable of handling composite terms in the Euclidean space. We will later show that the sequential analysis of our proposed algorithm also yields the same rate for dual extrapolation (Nesterov, 2007) in composite optimization, utilizing different proving techniques. As a result, we focus on the distributed optimization of composite SPP and propose FeDualEx.

## 3 PRELIMINARIES AND DEFINITIONS

We provide some preliminaries and definitions necessary for introducing FeDualEx. More details are included in Appendix C.1. To begin with, we lay out the notations.

**Notations.** We use $[n]$ to represent the set $\{1, 2, ..., n\}$. We use $\|\cdot\|$ to denote an arbitrary norm, $\|\cdot\|_*$ to denote the dual norm, and $\|\cdot\|_2$ to denote the Euclidean norm. We use $\nabla$ for gradients, $\partial$ for subgradients, and $\langle\cdot,\cdot\rangle$ for inner products. Related to the algorithm, we use English letters (e.g., $z, x, y$) to denote primal variables, Greek letters (e.g., $\omega, \varsigma, \mu, \nu$) to denote dual variables. We use $R$ for communication rounds, $K$ for local updates, $B$ for diameter bound, $G$ for gradient bound, $\beta$ for smoothness constant, $\sigma$ for standard deviation, $\xi$ for random samples. We use $h^*$ to denote the convex conjugate of a function $h$.

**Composite Saddle Point Optimization.**  We study composite saddle point optimization. Its objective is formally given in the following definition.

**Definition 1** (Composite SPP). *The objective of composite saddle point optimization is defined as*

$$\min_{x \in \mathcal{X}} \max_{y \in \mathcal{Y}} \phi(x,y) = f(x,y) + \psi_1(x) - \psi_2(y) \tag{1}$$

*where $f(x,y) = \frac{1}{M} \sum_{m=1}^{M} f_m(x,y)$ and $\psi_1(x)$, $\psi_2(y)$ are possibly non-smooth.*

It is typically evaluated by the duality gap: $\text{Gap}(\hat{x}, \hat{y}) = \max_{y \in \mathcal{Y}} \phi(\hat{x}, y) - \min_{x \in \mathcal{X}} \phi(x, \hat{y})$.

**Mirror Prox and Dual Extrapolation.**  Mirror prox (Nemirovski, 2004) and dual extrapolation (Nesterov, 2007) are classic methods for convex-concave SPP. Both are proximal algorithms based on the proximal operator defined as $\text{Prox}_{x'}^h(\cdot) = \arg\min_x \{\langle \cdot, x \rangle + V_{x'}^h(x)\}$, in which $V_{x'}^h(x) = h(x) - h(x') - \langle \nabla h(x'), x - x' \rangle$ is the Bregman divergence generated by some closed, strongly convex, and differentiable

$$x_t = \text{Prox}_{\bar{x}}^h(\mu_t)$$
$$x_{t+1/2} = \text{Prox}_{x_t}^h(\eta g(x_t))$$
$$\mu_{t+1} = \mu_t + \eta g(x_{t+1/2})$$

Figure 1: Dual Extrapolation.

function $h$. Both algorithms conduct two evaluations of the proximal operator, while dual extrapolation carries out updates in the dual space. Figure 1 gives a brief illustration of dual extrapolation with the proximal operator as in (Cohen et al., 2021), with details in Appendix C.1.

**Generalized Bregman Divergence.**  Recent advances in composite convex optimization (Yuan et al., 2021) have utilized the Generalized Bregman Divergence (Flammarion & Bach, 2017) for analyzing composite objectives. It incorporates the composite term into the distance-generating function of the vanilla Bregman divergence, and measures the distance in terms of one variable and the dual image of the other, with the key insight being the conjugate of a non-smooth generalized distance-generating function is differentiable.

**Definition 2** (Generalized Bregman Divergence (Flammarion & Bach, 2017)). *Generalized Bregman divergence is defined to be $\tilde{V}_{\mu'}^{h_t}(x) = h_t(x) - h_t(\nabla h_t^*(\mu')) - \langle \mu', x - \nabla h_t^*(\mu') \rangle$, where $h_t = h + t\eta\psi$ is a generalized distance-generating function that is closed and strongly convex, $t$ is the current number of iterations, $\eta$ is the step size, $h_t^*$ is the convex conjugate of $h_t$, and $\mu'$ is the dual image of $x'$, i.e., $\mu' \in \partial h_t(x')$ and $x' = \nabla h_t^*(\mu')$.*

Generalized Bregman divergence is suitable not only for non-smooth regularization but also for any convex constraints $\mathcal{C}$, taking $\psi(x) = 0$ if $x \in \mathcal{C}$ and $+\infty$ otherwise.

## 4 FEDERATED DUAL EXTRAPOLATION (FEDUALEX)

To tackle composite SPP in the DO paradigm, we acknowledge the challenges from several aspects. Specifically, the generality afforded by composite and/or saddle point problems results in a need for more sophisticated techniques that work with this additional structure. These concerns are further complicated by the challenges that arise for DO, where communication and aggregation need to be carefully handled under the distributed mechanism. In particular, Yuan et al. (2021) identified the "the curse of primal averaging" in composite federated optimization and advocated for dual aggregation. Dealing with these challenges altogether is rather non-trivial, as the techniques that are naturally suited for one would fail for another. In this regard, we first present FeDualEx (Algorithm 1) and several relevant novel definitions proposed for its adaptation to composite SPP. Then we analyze the convergence rate in the homogeneous setting.

### 4.1 THE FEDUALEX ALGORITHM

FeDualEx builds its core on the classic dual extrapolation, an extra-step algorithm geared for saddle point optimization. Its effectiveness has been widely verified in vanilla smooth convex-concave SPP. Furthermore, its updating sequence lies in the dual space which would naturally inherit the advantage of dual aggregation in composite federated optimization. The challenge remains for composite optimization, as relevant work is limited, and the existing composite extension for the extra-step method (He et al., 2015) is quite technically involved. Given that the smooth analysis of dual extrapolation is already non-trivial (Nesterov, 2007), no attempts were previously made for generalizing dual extrapolation to the composite optimization realm.

---

**Algorithm 1** FEDERATED-DUAL-EXTRAPOLATION (FeDualEx) for Composite SPP

---

**Input:** $\phi(z) = f(x,y) + \psi_1(x) - \psi_2(y) = \frac{1}{M}\sum_{m=1}^{M} f_m(x,y) + \psi_1(x) - \psi_2(y)$: objective function; $\ell(z)$: distance-generating function; $g_m(z) = (\nabla_x f_m(x,y), -\nabla_y f_m(x,y))$: gradient operator.

**Hyperparameters:** $R$: number of communication rounds; $K$: number of local update iterations; $\eta^s$: server step size; $\eta^c$: client step size.

**Dual Initialization:** $\varsigma_0 = 0$: initial dual variable, $\bar{\varsigma}$: fixed point in the dual space.

**Output:** Approximate solution $z = (x,y)$ to $\min_{x\in\mathcal{X}}\max_{y\in\mathcal{Y}}\phi(x,y)$

1: **for** $r = 0, 1, \ldots, R-1$ **do**
2:     Sample a subset of clients $C_r \subseteq [M]$
3:     **for** $m \in C_r$ **in parallel do**
4:         $\varsigma_{r,0}^m = \varsigma_r$
5:         **for** $k = 0, 1, \ldots, K-1$ **do**
6:             $z_{r,k}^m = \tilde{\text{Prox}}_{\bar{\varsigma}}^{\ell_{r,k}}(\varsigma_{r,k}^m)$       ▷ Two-step evaluation of the generalized proximal operator
7:             $z_{r,k+1/2}^m = \tilde{\text{Prox}}_{\bar{\varsigma}-\varsigma_{r,k}^m}^{\ell_{r,k+1}}(\eta^c g_m(z_{r,k}^m; \xi_{r,k}^m))$
8:             $\varsigma_{r,k+1}^m = \varsigma_{r,k}^m + \eta^c g_m(z_{r,k+1/2}^m; \xi_{r,k+1/2}^m)$       ▷ Dual variable update
9:         **end for**
10:     **end parallel for**
11:     $\Delta_r = \frac{1}{|\mathcal{C}_r|}\sum_{m\in\mathcal{C}_r}(\varsigma_{r,K}^m - \varsigma_{r,0}^m)$
12:     $\varsigma_{r+1} = \varsigma_r + \eta^s \Delta_r$       ▷ Server dual update
13: **end for**
14: **Return:** $\frac{1}{RK}\sum_{r=0}^{R-1}\sum_{k=0}^{K-1}\widehat{z_{r,k+1/2}}$ with $\widehat{z_{r,k+1/2}}$ defined in (4).

---

Inspired by recent advances in composite convex optimization, we recognize the Generalized Bregman Divergence (Flammarion & Bach, 2017) as a powerful tool for analyzing proximal methods for composite objectives. Adapting to the context of composite SPP, we make an extension to the Generalized Bregman Divergence for saddle functions, and provide the definition below.

**Definition 3** (Generalized Bregman Divergence for Saddle Functions). *The generalized distance-generating function for the optimization of* (1) *is* $\ell_t(z) = \ell(z) + t\eta\psi(z)$, *where* $\ell(z) = h_1(x) + h_2(y)$, $h_1$ *and* $h_2$ *are distance-generating functions for* $x$ *and* $y$, $\psi(z) = \psi_1(x) + \psi_2(y)$, $\eta$ *is the step size, and* $t$ *is the current number of iterations. It generates the following generalized Bregman divergence:*

$$\tilde{V}_{\varsigma'}^{\ell_t}(z) = \ell_t(z) - \ell_t(z') - \langle \varsigma', z - z'\rangle,$$

*where* $\varsigma'$ *is the preimage of* $z'$ *with respect to the gradient of the conjugate of* $\ell_t$, *i.e.,* $z' = \nabla\ell_t^*(\varsigma')$.

Yet as we notice in previous works (Flammarion & Bach, 2017; Yuan et al., 2021), generalized Bregman divergence is applied only for theoretical analysis. In terms of algorithm design, the previous proximal operator for composite convex optimization is based on the vanilla Bregman divergence plus the composite term, specifically, $\arg\min_x\{\langle\cdot, x\rangle + V_{x'}^h(x) + \eta\psi(x)\}$ in (Duchi et al., 2010; He et al., 2015), and $\arg\min_x\{\langle\cdot, x\rangle + h(x) + \eta t\psi(x)\}$ in (Xiao, 2010; Flammarion & Bach, 2017). However, we find this definition insufficient for dual extrapolation, as its dual update and the composite term from the extra step break certain parts of the analysis. In this effort, we propose a novel technical change to the proximal operator, directly replacing the Bregman divergence in the proximal operator with the generalized Bregman divergence.

**Definition 4** (Generalized Proximal Operator for Saddle Functions). *A proximal operation in the composite setting with generalized Bregman divergence for Saddle Functions is defined to be*

$$\tilde{\text{Prox}}_{\varsigma'}^{\ell_t}(g) := \arg\min_z\{\langle g, z\rangle + \tilde{V}_{\varsigma'}^{\ell_t}(z)\},$$

*where* $\varsigma'$ *is the dual image of* $z'$, *i.e.,* $z' = \nabla\ell_t^*(\varsigma')$, *and* $\varsigma' \in \partial\ell_t(z') = \nabla\ell(z') + \eta t\partial\psi(z')$.

Compared with the vanilla proximal operator in Section 3, this novel design for the composite adaptation of dual extrapolation is quite natural. It is different from previous proximal operators, which after expanding take the form $\arg\min_z\{\langle\cdot - \nabla\ell(z'), z\rangle + \ell_t(z)\}$ (Duchi et al., 2010) or $\arg\min_z\{\langle\cdot, z\rangle + \ell_t(z)\}$ (Xiao, 2010), whereas ours is $\tilde{\text{Prox}}_{\varsigma'}^h(\cdot) = \arg\min_z\{\langle\cdot - \varsigma', z\rangle + \ell_t(z)\}$.

These adaptations are necessary for technical reasons, as our algorithm involves prox operators on both the clients and the server to induce structure in the aggregated solution, which would otherwise

break the conventional analysis. (Specifically, using these previous notions yields extra composite terms in the analysis that do not cancel out but rather accumulate, thus hindering the convergence.)

With the novel definitions above, we are able to formally present FeDualEx in Algorithm 1. It follows the general structure of DO. For each client, the two-step evaluation of the generalized proximal operator and the final dual update are highlighted in green , which resembles the classic dual extrapolation updates in Figure 1. To align with our generalized proximal operator, we also move the primal initialization $\bar{x}$ in the original dual extrapolation to the dual space as $\bar{\varsigma}$. On the server, the dual variables from clients are aggregated first in the dual space, then projected to the primal with a mechanism later defined in (4).

## 4.2 CONVERGENCE ANALYSIS OF FEDUALEX

In this section, we provide the convergence analysis of FeDualEx for the homogeneous DO of composite SPP. We further assume the full participation of clients in each round for simplicity, but this condition can be trivially removed by lengthy analysis. We start by showing the equivalence between primal-dual projection and the generalized proximal operator, and for the convenience of analysis, reformulating the updating sequences with another pair of auxiliary dual variables.

**Projection Reformulation.** Generalized proximal operators can be presented as projections, i.e., the gradient of the conjugate of the generalized distance-generating function in Appendix C.2. Thus, line 6 to 8 in Algorithm 1 can be expanded by Definition 4, and rewrite as: (1) $z_{r,k}^m = \nabla\ell_{r,k}^*(\bar{\varsigma} - \varsigma_{r,k}^m)$; (2) $z_{r,k+1/2}^m = \nabla\ell_{r,k+1}^*((\bar{\varsigma} - \varsigma_{r,k}^m) - \eta^c g_m(z_{r,k}^m; \xi_{r,k}^m))$; (3) $\varsigma_{r,k+1}^m = \varsigma_{r,k}^m + \eta^c g_m(z_{r,k+1/2}^m; \xi_{r,k+1/2}^m)$.

Further define auxiliary dual variable $\omega_{r,k}^m = \bar{\varsigma} - \varsigma_{r,k}^m$. It satisfies immediately that $z_{r,k}^m = \nabla\ell_{r,k}^*(\omega_{r,k}^m)$, in which $\ell_{r,k}^*$ is the conjugate of $\ell_{r,k} = \ell + (\eta^s rK + k)\eta^c\psi$. And define $\omega_{r,k+1/2}^m$ to be the dual image of the intermediate variable $z_{r,k+1/2}^m$ such that $z_{r,k+1/2}^m = \nabla\ell_{r,k+1}^*(\omega_{r,k+1/2}^m)$. Then we get an equivalent updating sequence with the auxiliary dual variables.

$$\omega_{r,k+1/2}^m = \omega_{r,k}^m - \eta g_m(z_{r,k}^m; \xi_{r,k}^m), \qquad \omega_{r,k+1}^m = \omega_{r,k}^m - \eta g_m(z_{r,k+1/2}^m; \xi_{r,k+1/2}^m)$$

Define their average across clients, $\overline{\omega_{r,k}} = \frac{1}{M}\sum_{m=1}^M \omega_{r,k}^m$, $\overline{g_{r,k}} = \frac{1}{M}\sum_{m=1}^M g_m(z_{r,k}^m; \xi_{r,k}^m)$. Then we can analyze the following averaged dual shadow sequences:

$$\overline{\omega_{r,k+1/2}} = \overline{\omega_{r,k}} - \eta^c\overline{g_{r,k}}, \qquad (2) \qquad \overline{\omega_{r,k+1}} = \overline{\omega_{r,k}} - \eta^c\overline{g_{r,k+1/2}}. \qquad (3)$$

In the meantime, their shadow primal projections on the server are defined as

$$\widehat{z_{r,k}} = \nabla\ell_{r,k}^*(\overline{\omega_{r,k}}), \qquad \widehat{z_{r,k+1/2}} = \nabla\ell_{r,k+1}^*(\overline{\omega_{r,k+1/2}}). \qquad (4)$$

Next, we list the key assumptions. Detailed presentation and additional remarks that ease the understanding of proofs are also provided in Appendix C.3.

**Assumptions** *For the composite saddle function $\phi(x,y) = \frac{1}{M}\sum_{m=1}^M f_m(x,y) + \psi_1(x) - \psi_2(y)$, its gradient operator is given by $g = (\nabla_x f, -\nabla_y f)$ and $g = \frac{1}{M}\sum_{m=1}^M g_m$. We assume that*

    *a.(Convexity of $f$) $\forall m \in [M]$, $f_m(x,y)$ is convex in $x$ and concave in $y$.*
    *b.(Convexity of $\psi$) $\psi_1(x)$ is convex in $x$, and $\psi_2(y)$ is convex in $y$.*

    *c.(Lipschitzness of $g$) $g_m(z) = \begin{bmatrix} \nabla_x f_m(x,y) \\ -\nabla_y f_m(x,y) \end{bmatrix}$ is $\beta$-Lipschitz: $\|g_m(z) - g_m(z')\|_* \leq \beta\|z - z'\|$*

    *d.(Unbiased Estimate and Bounded Variance) $\forall m \in [M]$, for random sample $\xi^m$, $\mathbb{E}_\xi[g_m(z^m; \xi^m)] = g_m(z^m)$, and $\mathbb{E}_\xi[\|g_m(z^m; \xi^m) - g_m(z^m)\|_*^2] \leq \sigma^2$*
    *e.(Bounded Gradient) $\forall m \in [M]$, $\|g_m(z^m; \xi^m)\|_* \leq G$*
    *f. The distance-generating function $\ell$ is a Legendre function that is 1-strongly convex, i.e., $\forall z, z'$,*
$$\ell(z') - \ell(z) - \langle\nabla\ell(z), z' - z\rangle \geq \tfrac{1}{2}\|z' - z\|^2.$$

    *g.The optimization domain $\mathcal{Z}$ is compact w.r.t. Bregman divergence, i.e., $\forall z, z' \in \mathcal{Z}$, $V_{z'}^\ell(z) \leq B$.*

We would note that Assumption e (bounded gradient) is a standard assumption in classic distributed composite optimization (Duchi et al., 2011), and is made in other DO analysis (Stich, 2019; Li et al., 2020b; Yu et al., 2019; Yuan et al., 2021).

**Main Theorem.** Under the aforementioned assumptions, we present the following theorem that provides the convergence rate of FeDualEx in terms of the duality gap.

**Theorem 1** (Main). *Under [assumptions](#), the duality gap evaluated with the ergodic sequence generated by the intermediate steps of FeDualEx in Algorithm 1 is bounded by*

$$\mathbb{E}\Big[\operatorname{Gap}\Big(\frac{1}{RK}\sum_{r=0}^{R-1}\sum_{k=0}^{K-1}\widehat{z_{r,k+1/2}}\Big)\Big] \leq \frac{B}{\eta^c RK} + 20\beta^2(\eta^c)^3 K^2 G^2 + \frac{5\sigma^2\eta^c}{M} + 2^{\frac{3}{2}}\beta\eta^c KGB^{\frac{1}{2}}.$$

*Choosing step size* $\eta^c = \min\{\frac{1}{5^{\frac{1}{2}}\beta}, \frac{B^{\frac{1}{4}}}{20^{\frac{1}{4}}\beta^{\frac{1}{2}}G^{\frac{1}{2}}K^{\frac{3}{4}}R^{\frac{1}{4}}}, \frac{B^{\frac{1}{2}}M^{\frac{1}{2}}}{5^{\frac{1}{2}}\sigma R^{\frac{1}{2}}K^{\frac{1}{2}}}, \frac{B^{\frac{1}{4}}}{2^{\frac{3}{4}}\beta^{\frac{1}{2}}G^{\frac{1}{2}}KR^{\frac{1}{2}}}\},$

$$\mathbb{E}\Big[\operatorname{Gap}\Big(\frac{1}{RK}\sum_{r=0}^{R-1}\sum_{k=0}^{K-1}\widehat{z_{r,k+1/2}}\Big)\Big] \leq \frac{5^{\frac{1}{2}}\beta B}{RK} + \frac{20^{\frac{1}{4}}\beta^{\frac{1}{2}}G^{\frac{1}{2}}B^{\frac{3}{4}}}{K^{\frac{1}{4}}R^{\frac{3}{4}}} + \frac{5^{\frac{1}{2}}\sigma B^{\frac{1}{2}}}{M^{\frac{1}{2}}R^{\frac{1}{2}}K^{\frac{1}{2}}} + \frac{2^{\frac{3}{4}}\beta^{\frac{1}{2}}G^{\frac{1}{2}}B^{\frac{3}{4}}}{R^{\frac{1}{2}}}.$$

To the best of our knowledge, this is the first convergence rate for federated composite saddle point optimization. The $\mathcal{O}(\frac{1}{RK})$ and $\mathcal{O}(\frac{1}{\sqrt{MRK}})$ terms roughly match previous DO algorithms, where the noise term decays with the number of clients $M$. If $M$ is large enough, then the $\mathcal{O}(1/R^{\frac{1}{2}})$ term takes domination in terms of communication complexity. The convergence analysis further validates the effectiveness of FeDualEx, which then advances distributed optimization to a broad class of composite saddle point problems. The complete proof of Theorem 1 can be found in Appendix E.

**On Composite Convex Optimization.** We also analyze the convergence rate for FeDualEx under the federated composite convex optimization setting. As the following theorem shows, FeDualEx achieves the same $\mathcal{O}(1/R^{\frac{2}{3}})$ as in (Yuan et al., 2021). The proof is provided in Appendix F.

**Theorem 2.** *Under the convex counterparts of previous assumptions, choosing step size* $\eta^c = \min\{\frac{1}{5^{\frac{1}{2}}\beta}, \frac{B^{\frac{1}{4}}}{20^{\frac{1}{4}}\beta^{\frac{1}{2}}G^{\frac{1}{2}}K^{\frac{3}{4}}R^{\frac{1}{4}}}, \frac{B^{\frac{1}{2}}M^{\frac{1}{2}}}{5^{\frac{1}{2}}\sigma R^{\frac{1}{2}}K^{\frac{1}{2}}}, \frac{B^{\frac{1}{3}}}{2^{\frac{1}{3}}\beta^{\frac{1}{3}}G^{\frac{2}{3}}KR^{\frac{1}{3}}}\}$, *the ergodic intermediate sequence generated by FeDualEx for composite convex objectives satisfies*

$$\mathbb{E}\big[\phi\big(\frac{1}{RK}\sum_{r=0}^{R-1}\sum_{k=0}^{K-1}\widehat{x_{r,k+1/2}}\big) - \phi(x)\big] \leq \frac{5^{\frac{1}{2}}\beta B}{RK} + \frac{20^{\frac{1}{4}}\beta^{\frac{1}{2}}G^{\frac{1}{2}}B^{\frac{3}{4}}}{K^{\frac{1}{4}}R^{\frac{3}{4}}} + \frac{5^{\frac{1}{2}}\sigma B^{\frac{1}{2}}}{M^{\frac{1}{2}}R^{\frac{1}{2}}K^{\frac{1}{2}}} + \frac{2^{\frac{1}{3}}\beta^{\frac{1}{3}}G^{\frac{2}{3}}B^{\frac{2}{3}}}{R^{\frac{2}{3}}}.$$

Even though this rate is not preserved in composite saddle point optimization, we note that the optimization of SPP is much more general, and convexity itself is a stronger assumption. More specifically, the complicated setting, including the non-smooth term, the primal-dual projection, the extra-step saddle point optimization, etc., together limit the tools available for analysis.

**Remark On Heterogeneity.** Even for federated composite optimization (Yuan et al., 2021), the heterogeneous setting presents significant hurdles. Specifically, the involvement of heterogeneity is limited to quadratic functions, under which assumption the is gradient linear, and this simplifies the analysis. It further relies on the norm generated by its Hessian. For saddle functions, "quadraticity" (as well as a matrix-induced norm) is less well-defined, as the Jacobian of their gradient operator is not (symmetric) positive semidefinite in general. Such further advancements go beyond the scope of this paper. Thus, we regard Theorem 1 as a significant start for federated learning of composite SPP.

## 5 FEDUALEX IN SEQUENTIAL SETTINGS

**Stochastic Composite Saddle Point Optimization** FeDualEx can be naturally reduced to sequential stochastic optimization of composite SPP, which we term as *Sequential FeDualEx* or *Stochastic Dual Extrapolation*. By reducing the number of clients to one, thus eliminating the need for communication, the convergence analysis follows through smoothly and yields $\mathcal{O}(\frac{1}{\sqrt{T}})$ rate (denoting $K$ as $T$) expected for first-order stochastic algorithms. This is the first such rate in the non-Euclidean setting, matching the previous Euclidean rate (Mishchenko et al., 2020) and non-composite rate (Juditsky et al., 2011). Theorem 3 gives the result with proof in Appendix G.1.

**Theorem 3.** *Under the sequential versions of previous assumptions,* $\forall z \in \mathcal{Z}$, *choosing step size* $\eta = \min\{\frac{1}{3^{\frac{1}{2}}\beta}, \frac{B^{\frac{1}{2}}}{3^{\frac{1}{2}}\sigma T^{\frac{1}{2}}}\}$, *the ergodic intermediate sequence of stochastic dual extrapolation satisfies*

$$\mathbb{E}\big[\operatorname{Gap}(\frac{1}{T}\sum_{t=0}^{T-1}z_{t+1/2})\big] \leq \frac{3^{\frac{1}{2}}\beta B}{T} + \frac{3^{\frac{1}{2}}\sigma B^{\frac{1}{2}}}{T^{\frac{1}{2}}}.$$

$$\min_{\mathbf{x}\in\mathcal{X}} \max_{\mathbf{y}\in\mathcal{Y}}\langle\mathbf{Ax}-\mathbf{b},\mathbf{y}\rangle + \lambda\|\mathbf{x}\|_1 - \lambda\|\mathbf{y}\|_1$$

$$\mathbf{A}\in\mathbb{R}^{n\times m}, \quad \mathcal{X}=\{\mathbb{R}^m:\|\mathbf{x}\|_\infty\leq D\},$$

$$\mathbf{b}\in\mathbb{R}^n, \qquad \mathcal{Y}=\{\mathbb{R}^n:\|\mathbf{y}\|_\infty\leq D\}.$$

Figure 2: The composite SPP with $\ell_1$ regularization for sparsity (Jiang & Mokhtari, 2022).

$$\min_{\mathbf{X}\in\mathcal{X}} \max_{\mathbf{Y}\in\mathcal{Y}}\mathrm{Tr}\big((\mathbf{AX}-\mathbf{B})^\top\mathbf{Y}\big) + \lambda\|\mathbf{X}\|_* - \lambda\|\mathbf{Y}\|_*$$

$$\mathbf{A}\in\mathbb{R}^{n\times m}, \qquad \mathcal{X}=\{\mathbb{R}^{m\times p}:\|\mathbf{X}\|_2\leq D\},$$

$$\mathbf{B}\in\mathbb{R}^{n\times p}, \qquad \mathcal{Y}=\{\mathbb{R}^{n\times p}:\|\mathbf{Y}\|_2\leq D\}.$$

Figure 3: The composite SPP with nuclear norm low-rank regularization.

**Deterministic Composite Saddle Point Optimization** Further removing the noise in gradient, FeDualEx reduces to a deterministic algorithm for composite SPP. Even so, we are still generalizing the classic dual extrapolation algorithm to CO, and thus term the algorithm *Deterministic FeDualEx* or *Composite Dual Extrapolation*. Following a similar analysis, we are able to get the $\mathcal{O}(\frac{1}{T})$ rate as in previous work for CO (He et al., 2015) as well as the smooth dual extrapolation (Nesterov, 2007). The proof for Theorem 4 is in Appendix G.2, which is a much simpler one based on the recently proposed Relative Lipschitzness (Cohen et al., 2021).

**Theorem 4.** *Under the basic convexity assumption and $\beta$-Lipschitzness of $g$, $\forall z\in\mathcal{Z}$ and $\eta\leq\frac{1}{\beta}$, composite dual extrapolation satisfies* $\mathrm{Gap}(\frac{1}{T}\sum_{t=0}^{T-1}z_{t+1/2})\leq\frac{\beta B}{T}$.

## 6 EXPERIMENTS

To complement our largely theoretical results, we verify in this section the effectiveness of FeDualEx by numerical evaluation. Additional experiments and detailed settings are deferred to Appendix A.

**Composite Bilinear SPP.** We first test FeDualEx on composite bilinear problems with synthetic data. The problems considered are demonstrated in Figure 2 and 3, in which $m=600$, $n=300$, $p=20$, $\lambda=0.1$, $D=0.05$. The corresponding composite terms are $\ell_1$ regularization with $\ell_\infty$ ball constraint and nuclear regularization with spectral constraint. The purpose of $\ell_1$ regularization is to encourage sparsity and nuclear regularization to encourage a solution with low rank.

We compare FeDualEx against FedDualAvg, FedMiD (Yuan et al., 2021), and FedMiP proposed in Algorithm 2 in Appendix H. We note that methods like Extra Step Local SGD (Beznosikov et al., 2020) and SCAFFOLD-S (Karimireddy et al., 2020) are not suited to problems with non-smooth terms, but we include one of them for completeness, given that their rates are similar. For such a comparison, one can only compute the sub-gradient instead of the gradient (which does not everywhere exist). Projection needs to be applied as well to account for the constraints.

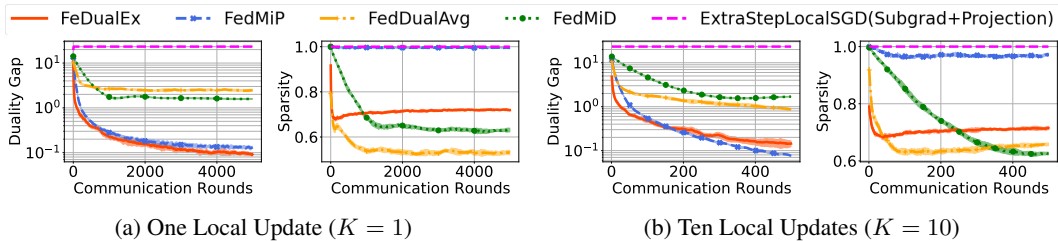

(a) One Local Update ($K=1$)    (b) Ten Local Updates ($K=10$)

Figure 4: Duality gap and sparsity of the solution for $\ell_1$ regularized SPP with $\ell_\infty$ constraint.

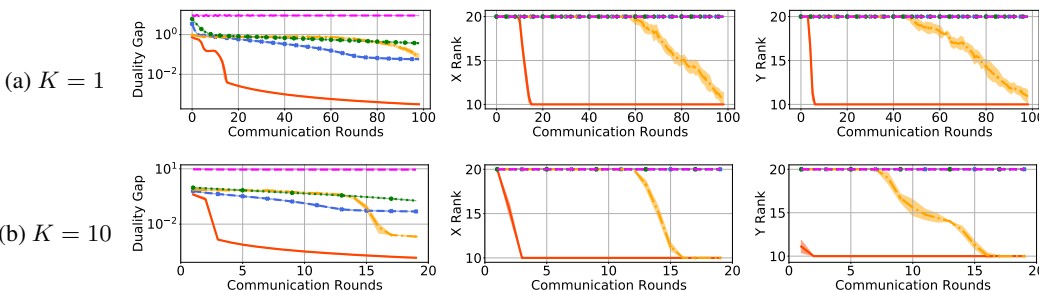

(a) $K=1$

(b) $K=10$

Figure 5: Duality gap and rank of the solution to the nuclear norm regularized SPP.

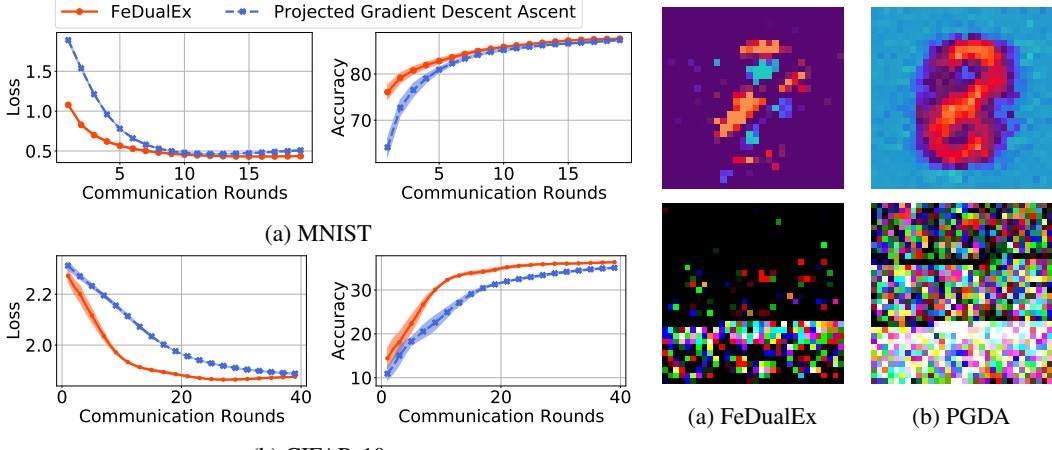

(a) MNIST

(b) CIFAR-10

Figure 6: Universal adversarial training loss and validation accuracy of logistic regression on unattacked data.

(a) FeDualEx  (b) PGDA

Figure 7: Attack generated from the universal-adversarially trained logistic regression on MNIST and CIFAR-10.

We evaluate the convergence in terms of the duality gap and also demonstrate the structure of the solution, i.e., sparsity or low-rankness. The duality gap of the problems of interest can be evaluated in closed form, which is derived in Appendix A.1 and A.2. The sparsity is measured by the ratio of non-zero entries to the parameter size, and we regard numbers less than $10^{-5}$ as zeros. For DO, we simulate $M = 100$ clients. For the gradient query of each client in each local update, we inject a Gaussian noise from $\mathcal{N}(0, \sigma^2)$, where $\sigma = 0.1$. The evaluation is conducted for two different settings: (a) $K = 1$ local update (b) $K = 10$ local updates. The results are demonstrated in Figure 4 and 5.

From the duality gap curves in Figure 4, we see that extra-step methods, i.e., FeDualEx and FedMiP converge to the order of $10^{-1}$ whereas FedDualAvg and FedMiD stay above $10^0$. Thus, it is evident that methods for composite convex optimization are no longer suited for composite saddle point optimization, and FeDualEx provides the first effective solution addressing the challenge. From the sparsity of the solution, we see that the dual methods demonstrate better adherence to regularization. Among the methods superior in saddle point optimization, FeDualEx reaches a sparsity of around 0.7 while FedMiP is around 0.95. This aligns with the previous analysis on the advantage of dual aggregation and further validates the effectiveness of FeDualEx for solving composite SPP. In addition, methods for smooth unconstrained optimization like ExtraStepLocalSGD do not converge for SPP with composite non-smooth terms, nor does it impose any desired structure, e.g., sparsity. We observe similar advantages of FeDualEx in convergence and inducing low-rankness from Figure 5 as well.

**Universal Adversarial Training of Logistic Regression.** We also consider the task of universal adversarial training (Shafahi et al., 2020) of logistic regression, i.e. the adversarial training against a universal adversarial perturbation (Moosavi-Dezfooli et al., 2017) targeted for all images in the dataset. In order to encourage the sparsity of the attack, we also impose an $l_1$ regularization on the attack. The problem formulation is given in Appendix A.3. We compare FeDualEx against direct aggregation of projected gradient descent ascent (PGDA) proposed in (Shafahi et al., 2020) Alg. 3.

We evaluate convergence with training loss, which is by no means an exact reflection of the duality gap. Still, we observe in Figure 6 that FeDualEx converges faster and delivers a better-hardened model with higher validation accuracy on unattacked data. Meanwhile, the vanilla aggregation of PGDA solutions yields a dense attack whereas FeDualEx achieves much better sparsity, as visualized in Figure 7. Furthermore, we observe that the attack generated by distributed PGDA is not only dense but also smoothed out to small values close to zero, averaged by the number of clients.

## 7 CONCLUSION AND FUTURE WORK

We advance distributed optimization to the broad class of composite SPP by proposing FeDualEx and providing, to our knowledge, the first convergence rate of its kind. We demonstrate the effectiveness of FeDualEx for inducing structures with empirical evaluation. We also show that the sequential version of FeDualEx provides a solution to composite stochastic saddle point optimization in the non-Euclidean setting. We recognize further study of the heterogeneous federated setting of composite saddle point optimization would be a challenging direction for future work.

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

# Appendices

In Appendix A, we provide details on experiment settings and additional experiments on the universal adversarial training of non-convex convolutional neural networks. In Appendix B, an extended literature review on various related subfields is included. Appendix C and D provide additional theoretical background, including relevant preliminaries, definitions, remarks, and technical lemmas. Appendix E, F, and G provide the convergence rates and complete proofs for FeDualEx in federated composite saddle point optimization, federated composite convex optimization, sequential stochastic composite optimization, and sequential deterministic composite optimization respectively. Finally, the algorithm of FedMiP is presented in Appendix H.

## A ADDITIONAL EXPERIMENTS AND SETUP DETAILS

### A.1 SETUP DETAILS FOR SADDLE POINT OPTIMIZATION WITH SPARSITY REGULARIZATION

We provide additional details for the SPP with the sparsity regularization demonstrated in the main text. We start by restating its formulation below:

$$\min_{\mathbf{x}\in\mathcal{X}} \max_{\mathbf{y}\in\mathcal{Y}} \langle \mathbf{A}\mathbf{x} - \mathbf{b}, \mathbf{y}\rangle + \lambda\|\mathbf{x}\|_1 - \lambda\|\mathbf{y}\|_1$$

$$\mathbf{A} \in \mathbb{R}^{n\times m}, \qquad \mathcal{X} = \{\mathbb{R}^m : \|\mathbf{x}\|_\infty \le D\},$$

$$\mathbf{b} \in \mathbb{R}^n, \qquad \mathcal{Y} = \{\mathbb{R}^n : \|\mathbf{y}\|_\infty \le D\}.$$

**Soft-Thresholding Operator for $\ell_1$ Norm Regularization.** By choosing the distance-generating function to be $\ell = \frac{1}{2}\|\mathbf{x}\|_2^2 + \frac{1}{2}\|\mathbf{y}\|_2^2$, the projection $\nabla\ell_{r,k}^*(\cdot)$ instantiates to the following element-wise soft-thresholding operator (Hastie et al., 2015; Jiang & Mokhtari, 2022):

$$T_{\lambda'}(\omega) := \begin{cases} 0 & \text{if } |\omega| \le \lambda' \\ (|\omega| - \lambda') \cdot \mathrm{sgn}(\omega) & \text{if } \lambda' < |\omega| \le \lambda' + D \ , \\ D \cdot \mathrm{sgn}(\omega) & \text{otherwise} \end{cases}$$

in which $\lambda' = \lambda\eta^c(\eta^s rK + k)$.

**Closed-Form Duality Gap.** The closed-form duality gap is given by

$$\mathrm{Gap}(\mathbf{x},\mathbf{y}) = D\|(|\mathbf{A}\mathbf{x} - \mathbf{b}| - \lambda)_+\|_1 + \lambda\|\mathbf{x}\|_1 + D\|(|\mathbf{A}^\top\mathbf{y}| - \lambda)_+\|_1 + \langle\mathbf{b},\mathbf{y}\rangle + \lambda\|\mathbf{y}\|_1,$$

where $|\cdot|$ and $()_+ = \max\{\cdot, 0\}$ are element-wise. We provide a brief derivation below. Since a constraint is equivalent to an indicator regularization, we move the $\ell_\infty$ constraint into the objective and denote $g_1(\cdot) = \|\cdot\|_1$, $g_2(\cdot) = \begin{cases} 0 & \text{if } \|\cdot\|_\infty \le D \\ \infty & \text{otherwise} \end{cases}$. By the definitions of duality gap in Definition 1 and convex conjugate in Definition 9, the duality gap equals to

$$\mathrm{Gap}(\mathbf{x},\mathbf{y}) = \max_{\mathbf{y}} \lambda\{\langle\frac{1}{\lambda}(\mathbf{A}\mathbf{x} - \mathbf{b}), \mathbf{y}\rangle - g_1(\mathbf{y}) - g_2(\mathbf{y}) + \|\mathbf{x}\|_1\}$$

$$- \min_{\mathbf{x}} \lambda\{\langle\frac{1}{\lambda}(\mathbf{A}^\top\mathbf{y}), \mathbf{x}\rangle + g_1(\mathbf{x}) + g_2(\mathbf{x}) - \|\mathbf{y}\|_1 - \frac{1}{\lambda}\mathbf{b}^\top\mathbf{y}\}$$

$$= \lambda(g_1 + g_2)^*(\frac{1}{\lambda}(\mathbf{A}\mathbf{x} - \mathbf{b})) + \lambda(g_1 + g_2)^*(\frac{1}{\lambda}(\mathbf{A}^\top\mathbf{y})) + \lambda\|\mathbf{x}\|_1 + \lambda\|\mathbf{y}\|_1 + \mathbf{b}^\top\mathbf{y}$$

$$= \inf_{\mathbf{u}}\{\lambda g_1^*(\mathbf{u}) + \lambda g_2^*(\frac{1}{\lambda}(\mathbf{A}\mathbf{x} - \mathbf{b}) - \mathbf{u})\} + \inf_{\mathbf{v}}\{\lambda g_1^*(\mathbf{v}) + \lambda g_2^*(\frac{1}{\lambda}(\mathbf{A}^\top\mathbf{y}) - \mathbf{v})\}$$

$$+ \lambda\|\mathbf{x}\|_1 + \lambda\|\mathbf{y}\|_1 + \mathbf{b}^\top\mathbf{y},$$

in which the last equality holds by Theorem 2.3.2, namely infimal convolution, in Chapter E of Hiriart-Urruty & Lemaréchal (2004). By definition of the convex conjugate, the convex conjugate of a norm $g(\cdot) = \|\cdot\|_p$ is defined to be $g^*(\cdot) = \begin{cases} 0 & \text{if } \|\cdot\|_q \le 1 \\ \infty & \text{otherwise} \end{cases}$, in which $\|\cdot\|_q$ is the dual norm of $\|\cdot\|_p$. Given that $\ell_1$ and $\ell_\infty$ are dual norms to each other, $g_1^*(\cdot) = \begin{cases} 0 & \text{if } \|\cdot\|_\infty \le 1 \\ \infty & \text{otherwise} \end{cases}$, $g_2^*(\cdot) = D\|\cdot\|_1$. Therefore the infimum is achieved when $\forall i \in [m], \forall j \in [n]$,

$$u_i = \begin{cases} \frac{1}{\lambda}(\mathbf{A}\mathbf{x} - \mathbf{b})_i & \text{if } |\frac{1}{\lambda}(\mathbf{A}\mathbf{x} - \mathbf{b})_i| \le 1 \\ \mathrm{sgn}(\frac{1}{\lambda}(\mathbf{A}\mathbf{x} - \mathbf{b})_i) & \text{otherwise} \end{cases}, \quad v_j = \begin{cases} \frac{1}{\lambda}(\mathbf{A}^\top\mathbf{y})_j & \text{if } |\frac{1}{\lambda}(\mathbf{A}^\top\mathbf{y})_j| \le 1 \\ \mathrm{sgn}(\frac{1}{\lambda}(\mathbf{A}^\top\mathbf{y})_j) & \text{otherwise} \end{cases},$$

which yields the closed-form duality gap.

**Additional Experiment Details.** We generate a fixed pair of $\mathbf{A}$ and $\mathbf{b}$ with each entry independently following the uniform distribution $\mathcal{U}_{[-1,1]}$. Each entry of the variables $\mathbf{x}$ and $\mathbf{y}$ is initialized independently from the distribution $\mathcal{U}_{[-D,D]}$. As in (Jiang & Mokhtari, 2022), we take $m = 600$, $n = 300$, $\lambda = 0.1$, $D = 0.05$. For DO, we simulate $M = 100$ clients. For the gradient query of each client

in each local update, we inject a Gaussian noise from $\mathcal{N}(0, \sigma^2)$. All $M = 100$ clients participate in each round; noise on each client is i.i.d. with $\sigma = 0.1$.

We only tune the global step size $\eta^s$ and the local step size $\eta^c$. For all experiments, the parameters are searched from the combination of $\eta^s \in \{1, 3e-1, 1e-1, 3e-2, 1e-2\}$ and $\eta^c \in \{1, 3e-1, 1e-1, 3e-2, 1e-2, 3e-3, 1e-3\}$. We run each setting for 10 different random seeds and report the mean and standard deviation in Figure 4.

## A.2 SADDLE POINT OPTIMIZATION WITH LOW-RANK REGULARIZATION

We test FeDualEx on the following SPP with nuclear norm regularization for low-rankness, in which we overuse the notation $\|\cdot\|_*$ for the matrix nuclear norm and $\|\cdot\|_2$ for the matrix spectral norm. We use $\mathrm{Tr}(\cdot)$ to denote the trace of a square matrix. And for the purpose of feasibility and convenience, we impose spectral norm constraints on the variables as well.

$$\min_{\mathbf{X} \in \mathcal{X}} \max_{\mathbf{Y} \in \mathcal{Y}} \mathrm{Tr}\big((\mathbf{AX} - \mathbf{B})^\top \mathbf{Y}\big) + \lambda \|\mathbf{X}\|_* - \lambda \|\mathbf{Y}\|_*$$

$$\mathbf{A} \in \mathbb{R}^{n \times m}, \qquad \mathcal{X} = \{\mathbb{R}^{m \times p} : \|\mathbf{X}\|_2 \leq D\},$$

$$\mathbf{B} \in \mathbb{R}^{n \times p}, \qquad \mathcal{Y} = \{\mathbb{R}^{n \times p} : \|\mathbf{Y}\|_2 \leq D\}.$$

**Soft-Thresholding Operator for Nuclear Norm Regularization.** By choosing the distance-generating function to be $\ell = \frac{1}{2}\|\mathbf{X}\|_F^2 + \frac{1}{2}\|\mathbf{Y}\|_F^2$ where $\|\cdot\|_F$ denotes the Frobenius norm, the projection $\nabla \ell^*_{r,k}(\cdot)$ instantiates to the following element-wise singular value soft-thresholding operator (Cai et al., 2010):

$$T_{\lambda'}(\mathbf{W}) := \mathbf{U} T_{\lambda'}(\mathbf{\Sigma}) \mathbf{V}^\top, \quad T_{\lambda'}(\mathbf{\Sigma}) = \mathrm{diag}(\mathrm{sgn}(\sigma_i(\mathbf{W})) \cdot \min\{\max\{\sigma_i(\mathbf{W}) - \lambda', 0\}, D\}),$$

in which $\lambda' = \lambda \eta^c (\eta^s r K + k)$, $\mathbf{W} = \mathbf{U} \mathbf{\Sigma} \mathbf{V}^\top$ is the singular value decomposition (SVD) of $\mathbf{W}$, and we overuse the notation $\sigma_i(\cdot)$ to represent the singular values.

**Closed-Form Duality Gap.** The closed-form duality gap is given by

$$\mathrm{Gap}(\mathbf{X}, \mathbf{Y}) = D\|\mathrm{diag}\big((|\sigma_i(\mathbf{AX} - \mathbf{B})| - \lambda)_+\big)\|_* + \lambda\|\mathbf{X}\|_*$$
$$+ D\|\mathrm{diag}\big((|\sigma_j(\mathbf{A}^\top \mathbf{Y})| - \lambda)_+\big)\|_* + \mathrm{Tr}\big(\mathbf{B}^\top \mathbf{Y}\big) + \lambda\|\mathbf{Y}\|_*,$$

We provide a brief derivation below. Since a constraint is equivalent to an indicator regularization, we move the spectral norm constraint into the objective and denote $g_1(\cdot) = \|\cdot\|_*$, $g_2(\cdot) = \begin{cases} 0 & \text{if } \|\cdot\|_2 \leq D \\ \infty & \text{otherwise} \end{cases}$. By the definitions of duality gap in Definition 1 and convex conjugate in Definition 9, the duality gap equals to

$$\mathrm{Gap}(\mathbf{X}, \mathbf{Y}) = \max_{\mathbf{Y}} \lambda\{\mathrm{Tr}\big(\frac{1}{\lambda}(\mathbf{AX} - \mathbf{B})^\top \mathbf{Y}\big) - g_1(\mathbf{Y}) - g_2(\mathbf{Y}) + \|\mathbf{X}\|_*\}$$

$$- \min_{\mathbf{X}} \lambda\{\{\mathrm{Tr}\big(\frac{1}{\lambda}(\mathbf{A}^\top \mathbf{Y})^\top \mathbf{X}\big) + g_1(\mathbf{X}) + g_2(\mathbf{X}) - \|\mathbf{Y}\|_* - \frac{1}{\lambda}\mathrm{Tr}\big(\mathbf{B}^\top \mathbf{Y}\big)\}$$

$$= \lambda(g_1 + g_2)^*\big(\frac{1}{\lambda}(\mathbf{AX} - \mathbf{B})\big) + \lambda(g_1 + g_2)^*\big(\frac{1}{\lambda}(\mathbf{A}^\top \mathbf{Y})\big)$$

$$+ \lambda\|\mathbf{X}\|_* + \lambda\|\mathbf{Y}\|_* + \mathrm{Tr}\big(\mathbf{B}^\top \mathbf{Y}\big)$$

$$= \inf_{\mathbf{P}}\{\lambda g_1^*(\mathbf{P}) + \lambda g_2^*\big(\frac{1}{\lambda}(\mathbf{AX} - \mathbf{B}) - \mathbf{P}\big)\} + \inf_{\mathbf{Q}}\{\lambda g_1^*(\mathbf{Q}) + \lambda g_2^*\big(\frac{1}{\lambda}(\mathbf{A}^\top \mathbf{Y}) - \mathbf{Q}\big)\}$$

$$+ \lambda\|\mathbf{X}\|_* + \lambda\|\mathbf{Y}\|_* + \mathrm{Tr}\big(\mathbf{B}^\top \mathbf{Y}\big),$$

in which the last equality holds by Theorem 2.3.2, namely infimal convolution, in Chapter E of Hiriart-Urruty & Lemaréchal (2004). By definition of the dual norm, we know that the nuclear norm and the spectral norm are dual norms to each other. Therefore, $g_1^*(\cdot) = \begin{cases} 0 & \text{if } \|\cdot\|_2 \leq 1 \\ \infty & \text{otherwise} \end{cases}$,

$g_2^*(\cdot) = D\|\cdot\|_*$. And the infimum is achieved when

$$\sigma_i(\mathbf{P}) = \begin{cases} \sigma_i\left(\frac{1}{\lambda}(\mathbf{Ax} - \mathbf{B})\right) & \text{if } |\sigma_i\left(\frac{1}{\lambda}(\mathbf{Ax} - \mathbf{B})\right)| \leq 1 \\ \text{sgn}\left(\sigma_i\left(\frac{1}{\lambda}(\mathbf{Ax} - \mathbf{B})\right)\right) & \text{otherwise} \end{cases},$$

$$\sigma_j(\mathbf{Q}) = \begin{cases} \sigma_j\left(\frac{1}{\lambda}(\mathbf{A}^\top\mathbf{y})\right) & \text{if } |\sigma_j\left(\frac{1}{\lambda}(\mathbf{A}^\top\mathbf{y})\right)| \leq 1 \\ \text{sgn}\left(\sigma_j\left(\frac{1}{\lambda}(\mathbf{A}^\top\mathbf{y})\right)\right) & \text{otherwise} \end{cases},$$

which yields the closed-form duality gap.

**Experiment Settings.** We generate a fixed pair of $\mathbf{A}$ and $\mathbf{B}$. Each entry of $\mathbf{A}$ and half of the columns in $\mathbf{B}$ follows the uniform distribution $\mathcal{U}_{[-1,1]}$ independently. Each entry of the variables $\mathbf{X}$ and $\mathbf{Y}$ is initialized independently from the distribution $\mathcal{U}_{[-1,1]}$. We take $m = 600$, $n = 300$, $p = 20$, $\lambda = 0.1$, $D = 0.05$. For DO, we simulate $M = 100$ clients. For the gradient query of each client in each local update, we inject a Gaussian noise from $\mathcal{N}(0, \sigma^2)$. All $M = 100$ clients participate in each round; noise on each client is i.i.d. with $\sigma = 0.1$. We only tune the global step size $\eta^s$ and the local step size $\eta^c$. For all experiments, the parameters are searched from the combination of $\eta^s \in \{1, 3e-1, 1e-1, 3e-2, 1e-2\}$ and $\eta^c \in \{10, 3, 1, 3e-1, 1e-1, 3e-2, 1e-2, 3e-3, 1e-3\}$. We run each setting for 10 different random seeds and plot the mean and the standard deviation.

We evaluate the convergence in terms of the duality gap and also demonstrate the rank of the solution, for both $\mathbf{X}$ and $\mathbf{Y}$. For the feasibility of low-rankness, we generate $\mathbf{B}$ to be of rank $\frac{p}{2}$, i.e. half of the columns of $B$ is linearly dependent on the other half. With $p = 20$, the optimal rank for the solution would most likely be 10. The evaluation is conducted for two different settings: (a) $K = 1$ local update for $R = 100$ rounds; (b) $K = 10$ local updates for $R = 20$ rounds. The results are demonstrated in Figure 5 correspondingly.

**Discussions.** From Figure 5, we can see that in the setting for low-rankness regularization, dual methods tend to perform better both in minimizing the duality gap and in encouraging a low-rank solution. In particular, FeDualEx, as a method geared for saddle point optimization, demonstrates better convergence in the duality gap than FedDualAvg. In the meantime, the solution given by FeDualEx quickly reaches the optimal rank of 10. This further reveals the potential of FeDualEx in coping with a variety of regularization and constraints.

## A.3 Universal Adversarial Training of Logistic Regression

We provide the problem formulation and detailed experiment setting for the universal adversarial training of logistic regression demonstrated in the main text.

**Problem Formulation.** As introduced, we impose an $l_1$ regularization on the attack to encourage sparsity in addition to the ball constraint. The problem can be formulated as the following SPP:

$$\min_{\mathbf{w} \in \mathbb{R}^d} \max_{\|\boldsymbol{\delta}\|_\infty \leq D} \frac{1}{n}\sum_{i=1}^n \ell(\mathbf{w}^\top(\mathbf{x}_i + \boldsymbol{\delta}), y_i) + \lambda\|\boldsymbol{\delta}\|_1$$

in which $\ell$ is the cross-entropy loss for multiclass logistic regression; $\mathbf{w} \in \mathbb{R}^d$ is the parameter; $\mathbf{x}_i \in \mathbb{R}^d$ is the data and $y_i$ is the label; $\boldsymbol{\delta} \in \mathbb{R}^d$ is the attack.

**Experiment Settings.** The training data for MNIST is evenly distributed across $M = 100$ clients, each possessing 600. The client makes $K = 5$ local updates and communicates for $R = 20$ rounds. For the CIFAR-10 experiments, each of the 100 clients holds 500 of the training data. The client makes $K = 5$ local updates and communicates for $R = 40$ rounds. $D = 0.05$ for data normalized between 0 and $\lambda = 0.1$. Validation is done on the whole validation dataset on the server with unattacked data. As before, the hyper-parameters are searched from the combination of $\eta^s \in \{1, 3e-1, 1e-1, 3e-2, 1e-2\}$ and $\eta^c \in \{10, 3, 1, 3e-1, 1e-1, 3e-2, 1e-2, 3e-3, 1e-3\}$. We run each setting for 10 different random seeds and plot the mean and the standard deviation in Figure 6.

**Attack Visualization.** The attack for MNIST has only one channel and is directly visualized with the color map from blue to red rescaled between the range of the attack, with blue being negative, red being positive, and purple being zero. The attack for CIFAR-10 contains 3 channels and can be directly visualized with RGB mode rescaled between 0 and 255. For the attack to be visible, we divide the value by its maximum then times the result by 4.

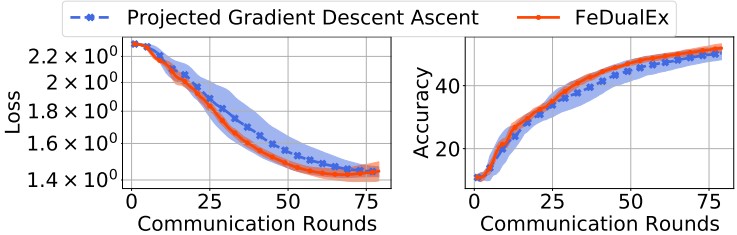

Figure 8: Training loss and validation accuracy of 3-layer CNN on unattacked data.

### A.4 UNIVERSAL ADVERSARIAL TRAINING OF NEURAL NETWORKS

Even though the theoretical result is derived with respect to convex functions, we experimentally demonstrate the convergence FeDualEx for non-convex functions with the adversarial training of neural networks on CIFAR-10. The model tested is a 3-layer convolutional neural network (CNN) with $16, 32$, and $64$ filters of size $3 \times 3$, each layer followed by a relu activation and a $2 \times 2$ max-pooling. The performance is demonstrated in Figure 8. The loss value is by no means an exact reflection of the duality gap, nevertheless, FeDualEx also converges for non-convex functions, yielding faster numerical convergence and better-hardened models in terms of validation accuracy on unattacked data. In addition, the sparsity of the attack generated by FeDualEx is $50.38\%$, whereas that by the vanilla distributed version of projected gradient descent ascent is $99.31\%$.

## B EXTENDED LITERATURE REVIEW

### B.1 DISTRIBUTED OPTIMIZATION / FEDERATED LEARNING

In recent years, distributed learning has received increasing attention in practice and theory. Earlier works in the field were known as "parallel" (Zinkevich et al., 2010) or "local" (Zhou & Cong, 2018; Stich, 2019), which are later recognized as the homogeneous case, where data across clients are assumed to be balanced and i.i.d. (independent and identically distributed), of federated learning (FL), specifically, Federated Averaging (FedAvg) (McMahan et al., 2017), DO or FL has been found appealing in various applications (Li et al., 2020a). On the theoretical front, Stich (2019) provides the first convergence rate for Local SGD, or, FedAvg under the homogeneous setting. **The distributed optimization paradigm we consider aligns with that in Local SGD (Stich, 2019).** The rate for LocalSGD has been improved with tighter analysis (Haddadpour et al., 2019; Khaled et al., 2020; Woodworth et al., 2020a; Glasgow et al., 2022) and acceleration techniques (Yuan & Ma, 2020; Mishchenko et al., 2022). Others also analyze FedAvg under heterogeneity (Haddadpour et al., 2019; Khaled et al., 2020; Woodworth et al., 2020b) and non-i.i.d. data (Li et al., 2020b) or in light propose improvements (Karimireddy et al., 2020). Recently, the idea of DO is further extended to higher-order methods (Bullins et al., 2021; Gupta et al., 2021; Safaryan et al., 2022). Due to the page limit, we refer the readers to (Cao et al., 2023; Wang et al., 2021; Kairouz et al., 2021) for more comprehensive reviews of DO and FL. In the meantime, we point out that none of the work mentioned above covers saddle point problems or non-smooth composite or constrained problems. For distributed saddle point optimization and federated composite optimization, we defer to the following subsections.

### B.2 SADDLE POINT OPTIMIZATION

The study of Saddle Point Optimization dates back to the very early gradient descent ascent (Arrow et al., 1958). It was later improved by the important ideas of extra-gradient (Korpelevich, 1976) and optimism (Popov, 1980). In light of these ideas, many algorithms were proposed for SPP (Solodov & Svaiter, 1999; Nemirovski, 2004; Nesterov, 2007; Chambolle & Pock, 2011; Mertikopoulos et al., 2019; Jiang & Mokhtari, 2022). Among them, in the convex-concave setting in particular, the most relevant and prominent ones are Nemirovski's mirror prox Nemirovski (2004) and Nesterov's dual extrapolation Nesterov (2007). They generalize respectively Mirror Descent (Nemirovskij & Yudin, 1983) and Dual Averaging (Nesterov, 2009) from convex optimization to monotone variational inequalities (VIs) which include SPP as one realization. Along with Tseng's Accelerated Proximal

Gradient (Tseng, 2008), they are the three methods that converge to an $\epsilon$-approximate solution in terms of duality gap at $\mathcal{O}(\frac{1}{T})$, the known best rate for a general convex-concave SPP (Ouyang & Xu, 2021; Lin et al., 2020). Mirror prox inspired many papers (Antonakopoulos et al., 2019; Chen et al., 2020) and is later extended to the stochastic setting (Juditsky et al., 2011; Mishchenko et al., 2020), the higher-order setting (Bullins & Lai, 2022), and even the composite setting (He et al., 2015), whose introduction we defer to the review of composite optimization. Dual extrapolation is later extended to non-monotone VIs (Song et al., 2020), yet its stochastic and composite versions are, to the best of our knowledge, not found. Kotsalis et al. (2022) recently studied optimal methods for stochastic variational inequalities, yet their result is limited to smooth VIs, not composite ones.

From the perspective of distributed optimization, several works have made preliminary progress for smooth and unconstrained SPP in the Euclidean space. Beznosikov et al. (2020) investigate the distributed extra-gradient method under various conditions and provide upper and lower bounds under strongly-convex strongly-concave and non-convex non-concave assumptions. Hou et al. (2021) propose FedAvg-S and SCAFFOLD-S based on FedAvg (McMahan et al., 2017) and SCAFFOLD (Karimireddy et al., 2020) for SPP, which achieves similar convergence rate to the distributed extra-gradient algorithm (Beznosikov et al., 2020) under the strong-convexity-concavity assumption. In addition, (Ramezani-Kebrya et al., 2023) studies the problem from the information compression perspective with the measure of communication bits. The topic of distributed or federated saddle point optimization is also found in recent applications of interest, e.g. adversarial domain adaptation (Shen et al., 2023). Yet, none of the existing works includes the study for SPP with constraints or composite possibly non-smooth regularization. Outside of our setting, Borodich et al. (2023) also studies composite SPP, but assumes composite terms to be smooth as well.

### B.3 COMPOSITE OPTIMIZATION

Composite optimization has been an important topic due to its reflection of real-world complexities. Representative works include composite mirror descent (Duchi et al., 2010) and regularized dual averaging (Xiao, 2010; Flammarion & Bach, 2017) that generalize mirror descent (Nemirovskij & Yudin, 1983) and dual averaging (Nesterov, 2009) in the context of composite convex optimization. Composite saddle point optimization, in comparison, appears dispersedly in early-day problems in practice (Buades et al., 2005; Aujol & Chambolle, 2005), often as a primal-dual reformulation of composite convex problems. Solving techniques such as smoothing (Nesterov, 2005) and primal-dual splitting (Combettes & Pesquet, 2012) were proposed, and numerical speed-ups were studied (He & Monteiro, 2015; 2016), while systematic convergence analysis on general composite SPP came later in time (He et al., 2015; Chambolle & Pock, 2016; Jiang & Mokhtari, 2022). Recently, Tominin et al. (2021); Borodich et al. (2022) also proposed acceleration techniques for composite SPP.

Most related among them, the pioneering composite mirror prox (CoMP) (He et al., 2015) constructs auxiliary variables for the composite regularization terms as an upper bound and thus moves the non-smooth term into the problem domain. Observing that the gradient operator for the auxiliary variable is constant, CoMP operates "as if" there were no composite components at all (He et al., 2015), and exhibits a $\mathcal{O}(\frac{1}{T})$ convergence rate that matches its smooth version (Nemirovski, 2004). In the stochastic setting, Mishchenko et al. (2020) analyzed a variant of stochastic mirror prox (Juditsky et al., 2011), which is then capable of handling composite terms in the Euclidean space. In this paper, we take a different approach that utilizes the generalized Bregman divergence and get the same rate for composite dual extrapolation.

For distributed composite optimization with local updates, Yuan et al. (2021) study Federated Mirror Descent, a natural extension of FedAvg that adapts to composite optimization under the convex setting. Along the way, they identified the "curse of primal averaging" specific to composite optimization in the DO paradigm, where the regularization-imposed structure on the client models may no longer hold after server primal averaging. To resolve this issue, they further proposed Federated Dual Averaging which brings the averaging step to the dual space. Tran Dinh et al. (2021) proposes a federated Douglas-Rachford splitting algorithm for nonconvex composite optimization. On the less related constrained optimization topic, Tong et al. (2020) proposed a federated learning algorithm for nonconvex sparse learning under $\ell_0$ constraint. To the best of our knowledge, the field of distributed optimization for composite SPP remains blank, which we regard as the main focus of this paper.

### B.4 Other Tangentially Related Work

**Decentralized Optimization.** Parallel to FL or DO with local updates, there is another line of work that studies *decentralized optimization* or *consensus optimization over networks*, in which machines communicate directly with each other based on their topological connectivity (Nedich et al., 2015). Classic algorithms mentioned previously are widely applied as well under this paradigm, for example, decentralized mirror descent (Rabbat, 2015) and decentralized (composite) dual averaging over networks (Duchi et al., 2011; Liu et al., 2022). Further in the context of composite optimization, Yan et al. (2023); Xiao et al. (2023) focus on composite non-convex objectives under the decentralized setting. Saddle point optimization has also been studied for decentralized optimization, including for proximal point-type methods (Liu et al., 2020) and extra-gradient methods (Rogozin et al., 2021; Beznosikov et al., 2021; 2022). In particular, Rogozin et al. (2021) studies decentralized "mirror prox" in the Euclidean space. We would like to point out that mirror prox in the Euclidean space reduces to vanilla extra-gradient methods. In addition, Aybat & Yazdandoost Hamedani (2016); Xu et al. (2021) study the saddle point reformulation for composite convex objectives over decentralized networks, which essentially focus on composite convex optimization. In the general context of distributed learning of composite SPP, by the judgment of the authors, we came across no paper in decentralized optimization similar to ours. More importantly, decentralized optimization focuses on topics like time-varying network topology (Kovalev et al., 2021a;b) or gossip schema (Dimakis et al., 2006), which are fundamentally different from our setting in terms of motivations, communication protocols, and techniques (Kairouz et al., 2021).

**Nonconvex-Nonconcave Saddle Point Problems.** For nonconvex-nonconcave SPP, several distributed learning methods have recently been proposed, including extra-gradient methods (Lee & Kim, 2021) and the Local Stochastic Gradient Descent Ascent (Local SGDA) (Sharma et al., 2022; 2023). Yet we emphasize that our object of analysis is composite SPP with possibly non-smooth regularization, and as remarked by Yuan et al. (2021), non-convex optimization for composite possibly non-smooth functions is in itself intricate even for sequential optimization, involving additional assumptions and sophisticated algorithm design (Li & Pong, 2015; Bredies et al., 2015), let alone distributed learning of SPP. Thus we focus on convex-concave analysis in this paper.

**Finite-Sum Optimization with Function Similarity.** Another line of work considers finite-sum optimization with function similarity, following the setting similar to DANE (Shamir et al., 2014). In this setting, each machine is assumed to maintain a fixed set of data so that the functions across machines can be $\delta$-similar with a high probability by large-sample concentration inequalities. In the context of distributed saddle points optimization, examples include (Kovalev et al., 2022) and (Beznosikov & Gasnikov, 2023). This setting is significantly different from ours because we do not consider $\delta$-similarity, and our optimization procedure is presented in an online scheme. In (Beznosikov & Gasnikov, 2023) in particular, local steps are also considered, but we would note that Beznosikov & Gasnikov (2023) require the server to take local steps instead of the clients, which also requires the presence of data on the server. This is done by making the first client the server and is in line with the setting in DANE (Shamir et al., 2014). In contrast, the server in our setting only aggregates the model and does not access any data, which also aligns with the privacy-preserving purpose in FL.

## C Additional Preliminaries, Definitions, and Remarks on Assumptions

In this section, we provide supplementary theoretical backgrounds for the algorithm and the convergence analysis of FeDualEx. We start by providing a more detailed introduction to the related algorithms, then list additional definitions necessary for the analysis. Before moving on to the main proof for FeDualEx, we state formally the assumptions made and provide additional remarks on the assumptions that better link them to their usage in the proof.

### C.1 Additional Preliminaries

To make this paper as self-contained as possible, in this section, we provide a brief overview of mirror descent, dual averaging, and their advancement in saddle point optimization, i.e., mirror prox and

dual extrapolation. More comprehensive introductions can be found in the original papers and in (Bubeck et al., 2015; Cohen et al., 2021). We slide into mirror descent from the simple and widely known projected gradient descent, namely vanilla gradient descent with constraint, therefore plus another projection of the updated sequence back to the feasible set.

### C.1.1 MIRROR DESCENT AND DUAL AVERAGING

We start by introducing projected gradient descent. Projected gradient descent first takes the gradient update, then projects the updated point back to the constraint by finding a feasible solution within the constraint that minimizes its Euclidean distance to the current point. The updating sequence is given below: $\forall t \in [T]$, $x_t \in \mathcal{X}$ whereas not necessarily for $x'_t$,

$$x'_{t+1} = x_t - \eta g(x_t)$$
$$x_{t+1} = \arg\min_{x \in \mathcal{X}} \frac{1}{2}\|x - x'_{t+1}\|_2^2.$$

**Mirror Descent (Nemirovskij & Yudin, 1983).** Mirror descent generalizes projected gradient descent to non-Euclidean space with the Bregman divergence (Bregman, 1967). We provide the definition of the Bregman divergence below.

**Definition 5** (Bregman Divergence (Bregman, 1967)). *Let $h : \mathbb{R}^d \to \mathbb{R} \cup \{\infty\}$ be a prox function or a distance-generating function that is closed, strictly convex, and differentiable in* **int dom** *$h$. The Bregman divergence for $x \in$* **dom** *$h$ and $y \in$* **int dom** *$h$ is defined to be*

$$V_y^h(x) = h(x) - h(y) - \langle \nabla h(y), x - y \rangle.$$

Mirror descent regards $\nabla h$ as a mirror map to the dual space, and follows the procedure below:

$$\nabla h(x'_{t+1}) = \nabla h(x_t) - \eta g(x_t)$$
$$x_{t+1} = \arg\min_{x \in \mathcal{X}} V_{x'_{t+1}}^h(x).$$

By choosing $h(\cdot) = \frac{1}{2}\|\cdot\|_2^2$ in the Euclidean space whose dual space is itself, mirror descent reduces to projected gradient descent.

Mirror descent can be presented from a proximal point of view, or in the online setting as in Beck & Teboulle (2003):

$$x_{t+1} = \arg\min_{x \in \mathcal{X}} \langle \eta g(x_t), x \rangle + V_{x_t}^h(x).$$

Such proximal operation with Bregman divergence is studied by others (Censor & Zenios, 1992), and is recently represented by a neatly defined proximal operator (Cohen et al., 2021).

**Definition 6** (Proximal Operator (Cohen et al., 2021)). *The Bregman divergence defined proximal operator is given by*

$$\text{Prox}_{x'}^h(\cdot) := \arg\min_{x \in \mathcal{X}}\{\langle \cdot, x \rangle + V_{x'}^h(x)\}.$$

In this spirit, the mirror descent algorithm can be written with one proximal operation:

$$x_{t+1} = \text{Prox}_{x_t}^h(\eta g(x_t)).$$

**Composite Mirror Descent (Duchi et al., 2010).** Mirror descent was later generalized to composite convex functions, i.e., the ones with regularization. The key modification is to include the regularization term in the proximal operator, yet not linearize the regularization term, since it could be non-smooth and thus non-differentiable. The updating sequence is given by

$$x_{t+1} = \arg\min_{x \in \mathcal{X}} \langle \eta g(x_t), x \rangle + V_{x_t}^h(x) + \eta \psi(x).$$

It can also be represented with a composite mirror map as in (Yuan et al., 2021):

$$x_{t+1} = \nabla(h + \eta\psi)^*(\nabla h(x_t) - \eta g(x_t)).$$

**Dual Averaging ([Nesterov, 2009](#)).** Compared with mirror descent, dual averaging moves the updating sequence to the dual space. The procedure of dual averaging is as follows ([Bubeck et al., 2015](#)):

$$\nabla h(x'_{t+1}) = \nabla h(x'_t) - \eta g(x_t)$$
$$x_{t+1} = \arg\min_{x \in \mathcal{X}} V^h_{x'_{t+1}}(x),$$

or equivalently as presented in ([Nesterov, 2009](#)) with the sequence of dual variables: $\forall t \in [T]$, $x_t \in \mathcal{X}, \mu_t \in \mathcal{X}^*$,

$$\mu_{t+1} = \mu_t - \eta g(x_t)$$
$$x_{t+1} = \nabla h^*(\mu_{t+1}).$$

This can be further simplified to

$$x_{t+1} = \arg\min_{x \in \mathcal{X}} \langle \eta \sum_{\tau=0}^{t} g(x_t), x \rangle + h(x).$$

**Composite Dual Averaging ([Xiao, 2010](#)).** Around the same time as composite mirror descent, composite dual averaging, also known as regularized dual averaging, was proposed with a similar idea of including the regularization term in the proximal operator. As presented in the original paper ([Xiao, 2010](#)):

$$x_{t+1} = \arg\min_{x \in \mathcal{X}} \langle \eta \sum_{\tau=0}^{t} g(x_\tau), x \rangle + \eta \beta_t h(x) + t\eta \psi(x),$$

in which $\{\beta_t\}_{t \geq 1}$ is a non-negative and non-decreasing input sequence. [Flammarion & Bach (2017)](#) adopted the case with constant sequence $\beta_t = \frac{1}{\eta}$,

$$x_{t+1} = \arg\min_{x \in \mathcal{X}} \langle \eta \sum_{\tau=0}^{t} g(x_\tau), x \rangle + h(x) + t\eta \psi(x),$$

and equivalently with composite mirror map:

$$\mu_{t+1} = \mu_t - \eta g(x_t)$$
$$x_{t+1} = \nabla(h + t\eta \psi)^*(\mu_{t+1}),$$

which is also presented in ([Yuan et al., 2021](#)).

### C.1.2 MIRROR PROX AND DUAL EXTRAPOLATION

**Mirror Prox ([Nemirovski, 2004](#)).** Mirror prox generalizes the extra-gradient method to non-Euclidean space as mirror descent compared with projected gradient descent. It was proposed for variational inequalities (VIs), including SPP. We first present the corresponding Bregman divergence in the saddle point setting, whose definition was not included in detail in ([Nemirovski, 2004](#)) but was later more clearly stated in ([Nesterov, 2007](#); [Shi et al., 2017](#)).

**Definition 7** (Bregman Divergence for Saddle Functions ([Nesterov, 2007](#))). *Let $\ell : \mathcal{X} \times \mathcal{Y} \to \mathbb{R} \cup \{\infty\}$ be a distance-generating function that is closed, strictly convex, and differentiable in* $\mathbf{int\,dom}\,\ell$*. For $z = (x, y) \in \mathcal{Z} = \mathcal{X} \times \mathcal{Y}$, the function and its gradient are defined as*

$$\ell(z) = h_1(x) + h_2(y), \qquad\qquad \nabla \ell(z) = \begin{bmatrix} \nabla_x h_1(x) \\ \nabla_y h_2(y) \end{bmatrix}.$$

*The Bregman divergence for $z = (x, y) \in \mathbf{dom}\,\ell$ and $z' = (x', y') \in \mathbf{int\,dom}\,\ell$ is defined to be*

$$V^\ell_{z'}(z) := \ell(z) - \ell(z') - \langle \nabla \ell(z'), z - z' \rangle.$$

*Notice that our notion of $\ell$ is not a saddle function, slightly different from that in [Shi et al. (2017)](#), but the Bregman divergence defined is the same as Eq. (6) in [Shi et al. (2017)](#) and Eq. (4.9) in [Nesterov (2007)](#).*

Mirror prox can also be viewed as an extra-step mirror descent. Most intuitively, by introducing an intermediate variable $z_{t+1/2}$, its procedure is as follows:

$$\nabla h(z'_{t+1/2}) = \nabla h(z_t) - \eta g(z_t)$$
$$z_{t+1/2} = \underset{z \in \mathcal{Z}}{\arg \min} \, V^h_{z'_{t+1/2}}(z)$$
$$\nabla h(z'_{t+1}) = \nabla h(z_t) - \eta g(z_{t+1/2})$$
$$z_{t+1} = \underset{z \in \mathcal{Z}}{\arg \min} \, V^h_{z'_{t+1}}(z).$$

And it can be represented with the proximal operator in Definition 6 as well. Following (Cohen et al., 2021), $\forall t \in [T]$, $z_t, z_{t+1/2} \in \mathcal{Z}$,

$$z_{t+1/2} = \text{Prox}^{\ell}_{z_t}(\eta g(z_t))$$
$$z_{t+1} = \text{Prox}^{\ell}_{z_t}(\eta g(z_{t+1/2})).$$

**Dual Extrapolation (Nesterov, 2007).** As in dual averaging, dual extrapolation moves the updating sequence of mirror prox to the dual space. Slightly different from a two-step dual averaging, dual extrapolation further initialize a fixed point in the primal space $\bar{z}$, and as presented in (Cohen et al., 2021), its procedure is as follows: $\forall t \in [T]$, $z_t, z_{t+1/2} \in \mathcal{Z}$, $\omega_t \in \mathcal{Z}^*$,

$$z_t = \text{Prox}^{\ell}_{\bar{z}}(\omega_t)$$
$$z_{t+1/2} = \text{Prox}^{\ell}_{z_t}(\eta g(z_t))$$
$$\omega_{t+1} = \omega_t + \eta g(z_{t+1/2}).$$

The updating sequence presented above is equivalent to that defined in the original paper (Nesterov, 2007), simply replacing the $\arg \max$ with $\arg \min$, and the dual variables with its additive inverse in the dual space.

## C.2 ADDITIONAL DEFINITIONS

In this subsection, we list additional definitions involved in the theoretical analysis in subsequent sections.

**Definition 8** (Legendre function (Rockafellar, 1970)). *A proper, convex, closed function $h : \mathbb{R}^d \to \mathbb{R} \cup \{\infty\}$ is called a Legendre function or a function of Legendre-type if (a) $h$ is strictly convex; (b) $h$ is essentially smooth, namely $h$ is differentiable on $\mathbf{int} \, \mathbf{dom} \, h$, and $\|\nabla h(x_t)\| \to \infty$ for every sequence $\{x_t\}_{t=0}^{\infty} \subset \mathbf{int} \, \mathbf{dom} \, h$ converging to a boundary point of $\mathbf{dom} \, h$ as $t \to \infty$.*

**Definition 9** (Convex Conjugate or Legendre–Fenchel Transformation (Boyd & Vandenberghe, 2004)). *The convex conjugate of a function $h$ is defined as*

$$h(s) = \sup_z \{\langle s, z \rangle - h(z)\}.$$

**Definition 10** (Differentiability of the conjugate of strictly convex function (Chapter E, Theorem 4.1.1 in Hiriart-Urruty & Lemaréchal (2004))). *For a strictly convex function $h$, $\mathbf{int} \, \mathbf{dom} \, h^* \neq \varnothing$ and $h^*$ is continuously differentiable on $\mathbf{int} \, \mathbf{dom} \, h^*$, with gradient defined as:*

$$\nabla h^*(s) = \underset{z}{\arg \min} \{\langle -s, z \rangle + h(z)\} \tag{5}$$

## C.3 FORMAL ASSUMPTIONS AND REMARKS

In this subsection, we state the assumptions formally and provide additional remarks that may help in understanding the theoretical analysis.

**Assumption 1** (Assumptions on the objective function). *For the composite saddle function $\phi(z) = f(x, y) + \psi_1(x) - \psi_2(y) = \frac{1}{M} \sum_{m=1}^{M} f_m(x, y) + \psi_1(x) - \psi_2(y)$, we assume that*

a. *(Local Convexity of $f$) $\forall m \in [M]$, $f_m(x, y)$ is convex in $x$ and concave in $y$.*

b. *(Convexity of $\psi$) $\psi_1(x)$ is convex in $x$, and $\psi_2(y)$ is convex in $y$.*

**Assumption 2** (Assumptions on the gradient operator). *For $f$ in the objective function, its gradient operator is given by $g = \begin{bmatrix} \nabla_x f \\ -\nabla_y f \end{bmatrix}$. By the linearity of gradient operators, $g = \frac{1}{M}\sum_{m=1}^{M} g_m$, and we assume that*

a. *(Local Lipschitzness of g) $\forall m \in [M]$, $g_m(z) = \begin{bmatrix} \nabla_x f_m(x,y) \\ -\nabla_y f_m(x,y) \end{bmatrix}$ is $\beta$-Lipschitz:*
$$\|g_m(z) - g_m(z')\|_* \leq \beta\|z - z'\|$$

b. *(Local Unbiased Estimate and Bounded Variance) For any client $m \in [M]$, the local gradient queried by some local random sample $\xi^m$ is unbiased and also bounded in variance, i.e., $\mathbb{E}_\xi[g_m(z^m;\xi^m)] = g_m(z^m)$, and*
$$\mathbb{E}_\xi\big[\|g_m(z^m;\xi^m) - g_m(z^m)\|_*^2\big] \leq \sigma^2$$

c. *(Bounded Gradient) $\forall m \in [M]$,*
$$\|g_m(z^m;\xi^m)\|_* \leq G$$

**Assumption 3** (Assumption on the distance-generating function). *The distance-generating function $h$ is a Legendre function that is 1-strongly convex, i.e., $\forall x, y$,*
$$h(y) - h(x) - \langle \nabla h(x), y - x\rangle \geq \frac{1}{2}\|y - x\|^2.$$

**Assumption 4.** *The domain of the optimization problem $\mathcal{Z}$ is compact in terms of Bregman Divergence, i.e., $\forall z, z' \in \mathcal{Z}$, $V_{z'}^\ell(z) \leq B$.*

**Remark 1.** *An immediate result of Assumption 1a is that, $\forall z = (x,y), z' = (x',y') \in \mathcal{Z}$*
$$f(x',y') - f(x,y') \leq \langle \nabla_x f(x',y'), x' - x\rangle,$$
$$f(x',y) - f(x',y') \leq \langle -\nabla_y f(x',y'), y' - y\rangle.$$

*Summing them up,*
$$f(x',y) - f(x,y') \leq \langle g(z'), z' - z\rangle.$$

**Remark 2.** *For any sequence of i.i.d. random variables $\xi_{0,0}^m, \xi_{0,1/2}^m, ..., \xi_{1,0}^m, \xi_{1,1/2}^m, ..., \xi_{r,k}^m, \xi_{r,k+1/2}^m$, let $\mathcal{F}_{r,k}$ denote the $\sigma$-field generated by the set $\{\xi_{j,t}^m : \forall m \in [M]$ and $((j = r, t \leq k)$ or $(j < r, k \in \{0, 1/2, ..., K-1, K-1/2\}))\}$. Then any $\xi_{r,k}^m$ is independent of $\mathcal{F}_{r,k-1/2}$, and Assumption 2b implies*
$$\mathbb{E}_{\mathcal{F}_{r,k}}\big[\|g_m(z_{r,k}^m;\xi_{r,k}^m) - g_m(z_{r,k}^m)\|_*^2 \mid \mathcal{F}_{r,k-1/2}\big] \leq \sigma^2.$$

**Remark 3** (Corollary 23.5.1. and Theorem 26.5. in Rockafellar (1970)). *For a closed convex (not necessarily differentiable) function $h$, $\partial h$ is the inverse of $\partial h^*$ in the sense of multi-valued mappings, i.e., $z \in \partial h^*(\varsigma)$ if and only if $\varsigma \in \partial h(z)$. Furthermore, if $h$ is of Legendre-type, meaning it is essentially strictly convex and essentially smooth, then $\partial h$ yields a well-defined $\nabla h$ that acts as a bijection, i.e., $(\nabla h)^{-1} = \nabla h^*$.*

**Remark 4.** *Assumption 3 and Remark 3 also trivially hold for $\ell$ from Definition 7 in the saddle point setting, and eventually, the generalized distance-generating function $\ell_t$ from Definition 3. Due to the strong convexity of $\ell_t$, $\nabla \ell_t^*$ is well-defined as noted in Definition 10. Together with the potential non-smoothness of $\ell_t$, Remark 3 implies that $z = \nabla \ell_t^*(\varsigma)$ if and only if $\varsigma \in \partial \ell_t(z)$.*

## D  ADDITIONAL TECHNICAL LEMMAS

In this section, we list some technical lemmas that are referenced in the proofs of the main theorem and its helping lemmas.

**Lemma 4** (Jensen's inequality). *For a convex function $\varphi(x)$, variables $x_1, ..., x_n$ in its domain, and positive weights $a_1, ..., a_n$,*
$$\varphi\Big(\frac{\sum_{i=1}^n a_i x_i}{\sum_{i=1}^n a_i}\Big) \leq \frac{\sum_{i=1}^n a_i \varphi(x_i)}{\sum_{i=1}^n a_i},$$

*and the inequality is reversed if $\varphi(x)$ is concave.*

**Lemma 5** (Cauchy-Schwarz inequality (Strang, 2006)). *For any $x$ and $y$ in an inner product space,*

$$\langle x, y \rangle \leq \|x\| \|y\|.$$

**Lemma 6** (Young's inequality (Lemma 1.45. in Sofonea & Matei (2009))). *Let $p, q \in \mathbb{R}$ be two conjugate exponents, that is $1 < p < \infty$, and $\frac{1}{p} + \frac{1}{q} = 1$. Then $\forall a, b \geq 0$,*

$$ab \leq \frac{a^p}{p} + \frac{b^q}{q}.$$

**Lemma 7** (AM-QM inequality). *For any set of positive integers $x_1, ..., x_n$,*

$$\Big( \sum_{i=1}^{n} x_i \Big)^2 \leq n \sum_{i=1}^{n} x_i^2. \tag{6}$$

**Lemma 8** (Lemma 2.3 in Jiang & Mokhtari (2022)). *Suppose Assumption 1 and 2 hold, then $\forall z = (x, y), z_1, ..., z_T \in \mathcal{Z}$ and $\theta_1, ..., \theta_T \geq 0$ with $\sum_{t=1}^{T} \theta_t = 1$, we have*

$$\phi(\sum_{t=1}^{T} \theta_t x_t, y) - \phi(x, \sum_{t=1}^{T} \theta_t y_t) \leq \sum_{t=1}^{T} \theta_t [\langle g(z_t), z_t - z \rangle + \psi(z_t) - \psi(z)],$$

*in which $\psi(z) = \psi_1(x) + \psi_2(y)$.*

*Proof.* For $\psi(z) = \psi_1(x) + \psi_2(y)$,

$$\begin{aligned}
\phi(x_t, y) - \phi(x, y_t) &= f(x_t, y) + \psi_1(x_t) - \psi_2(y) - f(x, y_t) - \psi_1(x) + \psi_2(y_t) \\
&= f(x_t, y) - f(x, y_t) + \psi(z_t) - \psi(z) \\
&\leq \langle g(z_t), z_t - z \rangle + \psi(z_t) - \psi(z),
\end{aligned}$$

where the inequality holds by convexity-concavity of $f(x, y)$, i.e. Remark 1. Then sum the inequality over $t = 1, ..., T$,

$$\sum_{t=1}^{T} \phi(\theta_t x_t, y) - \sum_{t=1}^{T} \phi(x, \theta_t y_t) \leq \sum_{t=1}^{T} \big[ \langle g(z_t), z_t - z \rangle + \psi(z_t) - \psi(z) \big].$$

Finally, by Jensen's inequality in Lemma 4,

$$\sum_{t=1}^{T} \phi(\theta_t x_t, y) \geq \phi\Big( \sum_{t=1}^{T} \theta_t x_t, y \Big), \qquad \sum_{t=1}^{T} \phi(x, \theta_t y_t) \leq \phi\Big( x, \sum_{t=1}^{T} \theta_t y_t \Big),$$

which completes the proof. □

**Lemma 9** (Theorem 4.2.1 in Hiriart-Urruty & Lemaréchal (2004)). *The conjugate of an $\alpha$-strongly convex function is $\frac{1}{\alpha}$-smooth. That is, for $h$ that is strongly convex with modulus $\alpha > 0$, $\forall x, x'$,*

$$\|\nabla h^*(x) - \nabla h^*(x')\| \leq \frac{1}{\alpha} \|x - x'\|.$$

**Lemma 10** (Lemma 2 in Flammarion & Bach (2017)). *Generalized Bregman divergence upper-bounds the Bregman divergence. That is, under Assumption 1 and 3, $\forall x \in \mathbf{dom}\, h$, $\forall \mu' \in \mathbf{int\, dom}\, h_t^*$ where $h_t = h + t\eta\psi$,*

$$\tilde{V}_{\mu'}^{h_t}(x) \geq V_{x'}^h(x),$$

*in which $x' = \nabla h_t^*(\mu')$.*

# E   COMPLETE ANALYSIS OF FEDUALEX FOR COMPOSITE SADDLE POINT PROBLEMS

We begin by reformulating the updating sequences with another pair of auxiliary dual variables. Expand the prox operator in Algorithm 1 line 6 to 8 by Definition 4, and rewrite by the gradient of the conjugate function in Definition 10,

$$z_{r,k}^m = \arg\min_z \{\langle \varsigma_{r,k}^m - \bar{\varsigma}, z\rangle + \ell_{r,k}(z)\} = \nabla\ell_{r,k}^*(\bar{\varsigma} - \varsigma_{r,k}^m)$$

$$z_{r,k+1/2}^m = \arg\min_z \{\langle \eta^c g_m(z_{r,k}^m; \xi_{r,k}^m) - (\bar{\varsigma} - \varsigma_{r,k}^m), z\rangle + \ell_{r,k+1}(z)\} = \nabla\ell_{r,k+1}^*((\bar{\varsigma} - \varsigma_{r,k}^m) - \eta^c g_m(z_{r,k}^m; \xi_{r,k}^m))$$

$$\varsigma_{r,k+1}^m = \varsigma_{r,k}^m + \eta^c g_m(z_{r,k+1/2}^m; \xi_{r,k+1/2}^m)$$

Define auxiliary dual variable $\omega_{r,k}^m = \bar{\varsigma} - \varsigma_{r,k}^m$. It satisfies immediately that $z_{r,k}^m = \nabla\ell_{r,k}^*(\omega_{r,k}^m)$, in which $\ell_{r,k}^*$ is the conjugate of $\ell_{r,k} = \ell + (\eta^s rK + k)\eta^c \psi$. And define $\omega_{r,k+1/2}^m$ to be the dual image of the intermediate variable $z_{r,k+1/2}^m$ such that $z_{r,k+1/2}^m = \nabla\ell_{r,k+1}^*(\omega_{r,k+1/2}^m)$. Then from the above updating sequence, we get an equivalent updating sequence for the auxiliary dual variables.

$$\omega_{r,k+1/2}^m = \omega_{r,k}^m - \eta g_m(z_{r,k}^m; \xi_{r,k}^m)$$
$$\omega_{r,k+1}^m = \omega_{r,k}^m - \eta g_m(z_{r,k+1/2}^m; \xi_{r,k+1/2}^m)$$

Now we analyze the following shadow sequences. Define

$$\overline{\omega_{r,k}} = \frac{1}{M}\sum_{m=1}^M \omega_{r,k}^m, \qquad\qquad \overline{g_{r,k}} = \frac{1}{M}\sum_{m=1}^M g_m(z_{r,k}^m; \xi_{r,k}^m),$$

then

$$\overline{\omega_{r,k+1/2}} = \overline{\omega_{r,k}} - \eta^c\overline{g_{r,k}}, \tag{2}$$
$$\overline{\omega_{r,k+1}} = \overline{\omega_{r,k}} - \eta^c\overline{g_{r,k+1/2}}. \tag{3}$$

In the meantime,

$$\widehat{z_{r,k}} = \nabla\ell_{r,k}^*(\overline{\omega_{r,k}}), \qquad\qquad \widehat{z_{r,k+1/2}} = \nabla\ell_{r,k+1}^*(\overline{\omega_{r,k+1/2}}). \tag{4}$$

## E.1   MAIN THEOREM AND PROOF

**Theorem 1** (Main). *Under assumptions, the duality gap evaluated with the ergodic sequence generated by the intermediate steps of FeDualEx in Algorithm 1 is bounded by*

$$\mathbb{E}\Big[\text{Gap}\Big(\frac{1}{RK}\sum_{r=0}^{R-1}\sum_{k=0}^{K-1}\widehat{z_{r,k+1/2}}\Big)\Big] \leq \frac{B}{\eta^c RK} + 20\beta^2(\eta^c)^3 K^2 G^2 + \frac{5\sigma^2\eta^c}{M} + 2^{\frac{3}{2}}\beta\eta^c KGB^{\frac{1}{2}}.$$

*Choosing step size* $\eta^c = \min\{\frac{1}{5^{\frac{1}{2}}\beta}, \frac{B^{\frac{1}{4}}}{20^{\frac{1}{4}}\beta^{\frac{1}{2}}G^{\frac{1}{2}}K^{\frac{3}{4}}R^{\frac{1}{4}}}, \frac{B^{\frac{1}{2}}M^{\frac{1}{2}}}{5^{\frac{1}{2}}\sigma R^{\frac{1}{2}}K^{\frac{1}{2}}}, \frac{B^{\frac{1}{4}}}{2^{\frac{3}{4}}\beta^{\frac{1}{2}}G^{\frac{1}{2}}KR^{\frac{1}{2}}}\},$

$$\mathbb{E}\Big[\text{Gap}\Big(\frac{1}{RK}\sum_{r=0}^{R-1}\sum_{k=0}^{K-1}\widehat{z_{r,k+1/2}}\Big)\Big] \leq \frac{5^{\frac{1}{2}}\beta B}{RK} + \frac{20^{\frac{1}{4}}\beta^{\frac{1}{2}}G^{\frac{1}{2}}B^{\frac{3}{4}}}{K^{\frac{1}{4}}R^{\frac{3}{4}}} + \frac{5^{\frac{1}{2}}\sigma B^{\frac{1}{2}}}{M^{\frac{1}{2}}R^{\frac{1}{2}}K^{\frac{1}{2}}} + \frac{2^{\frac{3}{4}}\beta^{\frac{1}{2}}G^{\frac{1}{2}}B^{\frac{3}{4}}}{R^{\frac{1}{2}}}.$$

*Proof.* The proof of the main theorem relies on Lemma 1, the bound for the non-smooth term, and Lemma 2, the bound for the smooth term. These two lemmas are combined in Lemma 3 and then yield the per-step progress for FeDualEx. The three lemmas are listed and proved right after this theorem. Here, we finish proving the main theorem from the per-step progress.

Starting from Lemma 3, we telescope for all local updates $k \in \{0, ..., K-1\}$ after the same communication round $r$.

$$\eta^c \mathbb{E}\Big[ \sum_{k=0}^{K-1} \big[ \langle g(\widehat{z_{r,k+1/2}}), \widehat{z_{r,k+1/2}} - z \rangle + \psi(\widehat{z_{r,k+1/2}}) - \psi(z) \big] \Big]$$

$$\leq \tilde{V}_{\omega_{r,0}}^{\ell_{r,0}}(z) - \tilde{V}_{\omega_{r,K}}^{\ell_{r,K}}(z) + \frac{5\sigma^2(\eta^c)^2 K}{M} + 20 \sum_{k=0}^{K-1} \beta^2(\eta^c)^4(k+1)^2 G^2 + 2^{\frac{3}{2}} \sum_{k=0}^{K-1} \beta(\eta^c)^2(k+1)GB^{\frac{1}{2}}$$

$$\leq \tilde{V}_{\omega_{r,0}}^{\ell_{r,0}}(z) - \tilde{V}_{\omega_{r,K}}^{\ell_{r,K}}(z) + \frac{5\sigma^2(\eta^c)^2 K}{M} + 20 \sum_{k=0}^{K-1} \beta^2(\eta^c)^4 K^2 G^2 + 2^{\frac{3}{2}} \sum_{k=0}^{K-1} \beta(\eta^c)^2 KGB^{\frac{1}{2}}$$

$$\leq \tilde{V}_{\omega_{r,0}}^{\ell_{r,0}}(z) - \tilde{V}_{\omega_{r,K}}^{\ell_{r,K}}(z) + \frac{5\sigma^2(\eta^c)^2 K}{M} + 20\beta^2(\eta^c)^4 K^3 G^2 + 2^{\frac{3}{2}} \beta(\eta^c)^2 K^2 GB^{\frac{1}{2}}.$$

As we initialize the local dual updates on all clients after each communication with the dual average of the previous round's last update, $\forall r \in \{1, ..., R\}$, the first variable in this round $\overline{\omega_{r,0}}$ is the same as the last variable $\overline{\omega_{r-1,0}}$ in the previous round. As a result, taking the server step size $\eta^s = 1$, we can further telescope across all rounds and have

$$\eta^c \mathbb{E}\Big[ \sum_{r=0}^{R-1} \sum_{k=0}^{K-1} \big[ \langle g(\widehat{z_{r,k+1/2}}), \widehat{z_{r,k+1/2}} - z \rangle + \psi(\widehat{z_{r,k+1/2}}) - \psi(z) \big] \Big]$$

$$\leq \tilde{V}_{\omega_{0,0}}^{\ell_{0,0}}(z) - \tilde{V}_{\omega_{R,K}}^{\ell_{R,K}}(z) + \frac{5\sigma^2(\eta^c)^2 KR}{M} + 20\beta^2(\eta^c)^4 K^3 RG^2 + 2^{\frac{3}{2}} \beta(\eta^c)^2 K^2 RGB^{\frac{1}{2}}.$$

Notice that the generalized Bregman divergence $\tilde{V}_{\omega_{0,0}}^{\ell_{0,0}}(z) = \tilde{V}_{\bar{\varsigma} - \varsigma_0}^{\ell_{0,0}}(z) = \tilde{V}_{\bar{\varsigma}}^{\ell}(z) = V_{z_0}^{\ell}(z)$, where $z_0 = \nabla \ell^*(\bar{\varsigma})$. Thus, by Assumption 4, $\tilde{V}_{\omega_{0,0}}^{\ell_{0,0}}(z) \leq B$. Dividing $\eta^c KR$ on both sides of the equation, we get

$$\eta^c \mathbb{E}\Big[ \frac{1}{RK} \sum_{r=0}^{R-1} \sum_{k=0}^{K-1} \big[ \langle g(\widehat{z_{r,k+1/2}}), \widehat{z_{r,k+1/2}} - z \rangle + \psi(\widehat{z_{r,k+1/2}}) - \psi(z) \big] \Big]$$

$$\leq \frac{B}{\eta^c RK} + \frac{5\sigma^2 \eta^c}{M} + 20\beta^2(\eta^c)^3 K^2 G^2 + 2^{\frac{3}{2}} \beta \eta^c KGB^{\frac{1}{2}}.$$

Finally, applying Lemma 8 completes the proof. $\qquad \square$

**Lemma 1** (Bounding the Regularization Term). $\forall z$,

$$\eta^c \big[ \psi(\widehat{z_{r,k+1/2}}) - \psi(z) \big] = \tilde{V}_{\omega_{r,k}}^{\ell_{r,k}}(z) - \tilde{V}_{\omega_{r,k+1}}^{\ell_{r,k+1}}(z) - \tilde{V}_{\omega_{r,k}}^{\ell_{r,k}}(\widehat{z_{r,k+1/2}}) - \tilde{V}_{\omega_{r,k+1/2}}^{\ell_{r,k+1}}(\widehat{z_{r,k+1}})$$
$$+ \eta^c \langle \overline{g_{r,k+1/2}} - \overline{g_{r,k}}, \widehat{z_{r,k+1/2}} - \widehat{z_{r,k+1}} \rangle + \eta^c \langle \overline{g_{r,k+1/2}}, z - \widehat{z_{r,k+1/2}} \rangle$$

*Proof.* By the definition of generalized Bregman divergence and the updating sequence in Eq. (2), $\forall z$,

$$\tilde{V}_{\omega_{r,k+1/2}}^{\ell_{r,k+1}}(z) = \ell_{r,k+1}(z) - \ell_{r,k+1}(\widehat{z_{r,k+1/2}}) - \langle \overline{\omega_{r,k+1/2}}, z - \widehat{z_{r,k+1/2}} \rangle$$

$$= \ell_{r,k+1}(z) - \ell_{r,k+1}(\widehat{z_{r,k+1/2}}) - \langle \overline{\omega_{r,k}} - \eta^c \overline{g_{r,k}}, z - \widehat{z_{r,k+1/2}} \rangle$$

$$= \ell_{r,k}(z) - \ell_{r,k}(\widehat{z_{r,k+1/2}}) + \eta^c \big[ \psi(z) - \psi(\widehat{z_{r,k+1/2}}) \big]$$
$$- \langle \overline{\omega_{r,k}}, z - \widehat{z_{r,k+1/2}} \rangle + \eta^c \langle \overline{g_{r,k}}, z - \widehat{z_{r,k+1/2}} \rangle. \tag{7}$$

Similarly, we can have for the updating sequence in Eq. (3) that $\forall z$,

$$\tilde{V}_{\omega_{r,k+1}}^{\ell_{r,k+1}}(z) = \ell_{r,k}(z) - \ell_{r,k}(\widehat{z_{r,k+1}}) + \eta^c \big[ \psi(z) - \psi(\widehat{z_{r,k+1}}) \big]$$
$$- \langle \overline{\omega_{r,k}}, z - \widehat{z_{r,k+1}} \rangle + \eta^c \langle \overline{g_{r,k+1/2}}, z - \widehat{z_{r,k+1}} \rangle. \tag{8}$$

Plug $z = \widehat{z_{r,k+1}}$ into Eq. (7),

$$\tilde{V}_{\omega_{r,k+1/2}}^{\ell_{r,k+1}}(\widehat{z_{r,k+1}}) = \ell_{r,k}(\widehat{z_{r,k+1}}) - \ell_{r,k}(\widehat{z_{r,k+1/2}}) + \eta^c \big[ \psi(\widehat{z_{r,k+1}}) - \psi(\widehat{z_{r,k+1/2}}) \big]$$
$$- \langle \overline{\omega_{r,k}}, \widehat{z_{r,k+1}} - \widehat{z_{r,k+1/2}} \rangle + \eta^c \langle \overline{g_{r,k}}, \widehat{z_{r,k+1}} - \widehat{z_{r,k+1/2}} \rangle.$$

Add this up with Eq. (8),

$$\tilde{V}^{\ell_{r,k+1}}_{\omega_{r,k+1/2}}(\widehat{z_{r,k+1}}) + \tilde{V}^{\ell_{r,k+1}}_{\omega_{r,k+1}}(z) = \underbrace{\ell_{r,k}(z) - \ell_{r,k}(\widehat{z_{r,k+1/2}}) - \langle\overline{\omega_{r,k}}, z - \widehat{z_{r,k+1/2}}\rangle}_{A1}$$

$$+ \eta^c\big[\psi(z) - \psi(\widehat{z_{r,k+1/2}})\big]$$

$$+ \underbrace{\eta^c\langle\overline{g_{r,k}}, \widehat{z_{r,k+1}} - \widehat{z_{r,k+1/2}}\rangle + \eta^c\langle\overline{g_{r,k+1/2}}, z - \widehat{z_{r,k+1}}\rangle}_{A2}.$$

For $A1$ we have

$$A1 = \ell_{r,k}(z) - \ell_{r,k}(\widehat{z_{r,k}}) - \langle\overline{\omega_{r,k}}, z - \widehat{z_{r,k}}\rangle - \ell_{r,k}(\widehat{z_{r,k+1/2}}) + \ell_{r,k}(\widehat{z_{r,k}}) + \langle\overline{\omega_{r,k}}, \widehat{z_{r,k+1/2}} - \widehat{z_{r,k}}\rangle$$

$$= \tilde{V}^{\ell_{r,k}}_{\omega_{r,k}}(z) - \tilde{V}^{\ell_{r,k}}_{\omega_{r,k}}(\widehat{z_{r,k+1/2}}).$$

For $A2$ we have

$$A2 = \eta^c\langle\overline{g_{r,k}}, \widehat{z_{r,k+1}} - \widehat{z_{r,k+1/2}}\rangle + \eta^c\langle\overline{g_{r,k+1/2}}, \widehat{z_{r,k+1/2}} - \widehat{z_{r,k+1}}\rangle + \eta^c\langle\overline{g_{r,k+1/2}}, z - \widehat{z_{r,k+1/2}}\rangle$$

$$= \eta^c\langle\overline{g_{r,k+1/2}}, z - \widehat{z_{r,k+1/2}}\rangle + \eta^c\langle\overline{g_{r,k+1/2}} - \overline{g_{r,k}}, \widehat{z_{r,k+1/2}} - \widehat{z_{r,k+1}}\rangle$$

Plug $A1$ and $A2$ back in completes the proof. $\qquad\square$

For the purpose of clarity, we demonstrate how we generate the terms to be separately bounded for the smooth part with the following Lemma 2, which holds trivially by the linearity of the gradient operator $g = \frac{1}{M}\sum_{m=1}^{M} g_m$ and then direct cancellation.

**Lemma 2** (Bounding the Smooth Term). $\forall z$,

$$\langle g(\widehat{z_{r,k+1/2}}), \widehat{z_{r,k+1/2}} - z\rangle = \langle\overline{g_{r,k+1/2}}, \widehat{z_{r,k+1/2}} - z\rangle + \langle\frac{1}{M}\sum_{m=1}^{M} g_m(z^m_{r,k+1/2}) - \overline{g_{r,k+1/2}}, \widehat{z_{r,k+1/2}} - z\rangle$$

$$+ \langle\frac{1}{M}\sum_{m=1}^{M}[g_m(\widehat{z_{r,k+1/2}}) - g_m(z^m_{r,k+1/2})], \widehat{z_{r,k+1/2}} - z\rangle$$

Based on the previous two lemmas, we arrive at the following lemma that bounds the per-step progress of FeDualEx.

**Lemma 3** (Per-step Progress for FeDualEx in Saddle Point Setting). *For $\eta^c \le \frac{1}{5^{\frac{1}{2}}\beta}$,*

$$\eta^c\mathbb{E}\big[\langle g(\widehat{z_{r,k+1/2}}), \widehat{z_{r,k+1/2}} - z\rangle + \psi(\widehat{z_{r,k+1/2}}) - \psi(z)\big]$$

$$\le \tilde{V}^{\ell_{r,k}}_{\omega_{r,k}}(z) - \tilde{V}^{\ell_{r,k+1}}_{\omega_{r,k+1}}(z) + \frac{5\sigma^2(\eta^c)^2}{M} + 20\beta^2(\eta^c)^4(k+1)^2 G^2 + 2^{\frac{3}{2}}\beta(\eta^c)^2(k+1)GB^{\frac{1}{2}}.$$

*Proof.* Based on the previous two lemmas, we can get the following simply by summing them up, in which we denote the left-hand side as LHS for simplicity.

$$\text{LHS} := \eta^c\big[\langle g(\widehat{z_{r,k+1/2}}), \widehat{z_{r,k+1/2}} - z\rangle + \psi(\widehat{z_{r,k+1/2}}) - \psi(z)\big]$$

$$\le \tilde{V}^{\ell_{r,k}}_{\omega_{r,k}}(z) - \tilde{V}^{\ell_{r,k+1}}_{\omega_{r,k+1}}(z) \underbrace{- \tilde{V}^{\ell_{r,k}}_{\omega_{r,k}}(\widehat{z_{r,k+1/2}}) - \tilde{V}^{\ell_{r,k+1}}_{\omega_{r,k+1/2}}(\widehat{z_{r,k+1}})}_{A3}$$

$$+ \eta^c\langle\overline{g_{r,k+1/2}} - \overline{g_{r,k}}, \widehat{z_{r,k+1/2}} - \widehat{z_{r,k+1}}\rangle$$

$$+ \eta^c\langle\frac{1}{M}\sum_{m=1}^{M} g_m(z^m_{r,k+1/2}) - \overline{g_{r,k+1/2}}, \widehat{z_{r,k+1/2}} - z\rangle$$

$$+ \eta^c\langle\frac{1}{M}\sum_{m=1}^{M}[g_m(\widehat{z_{r,k+1/2}}) - g_m(z^m_{r,k+1/2})], \widehat{z_{r,k+1/2}} - z\rangle$$

For the two generalized Bregman divergence terms in $A3$, we bound them by Lemma 10 and the strong convexity of $\ell$ in Remark 4,

$$A3 \leq -V^{\ell}_{\widehat{z_{r,k}}}(\widehat{z_{r,k+1/2}}) - V^{\ell}_{\widehat{z_{r,k+1/2}}}(\widehat{z_{r,k+1}})$$
$$\leq -\frac{1}{2}\|\widehat{z_{r,k}} - \widehat{z_{r,k+1/2}}\|^2 - \frac{1}{2}\|\widehat{z_{r,k+1/2}} - \widehat{z_{r,k+1}}\|^2$$

As a result,

$$\text{LHS} \leq \tilde{V}^{\ell_{r,k}}_{\omega_{r,k}}(z) - \tilde{V}^{\ell_{r,k+1}}_{\omega_{r,k+1}}(z) - \frac{1}{2}\|\widehat{z_{r,k}} - \widehat{z_{r,k+1/2}}\|^2$$
$$\underbrace{-\frac{1}{2}\|\widehat{z_{r,k+1/2}} - \widehat{z_{r,k+1}}\|^2 + \eta^c\langle \overline{g_{r,k+1/2}} - \overline{g_{r,k}}, \widehat{z_{r,k+1/2}} - \widehat{z_{r,k+1}}\rangle}_{A4}$$
$$+ \eta^c\langle \frac{1}{M}\sum_{m=1}^{M} g_m(z^m_{r,k+1/2}) - \overline{g_{r,k+1/2}}, \widehat{z_{r,k+1/2}} - z\rangle$$
$$+ \eta^c\langle \frac{1}{M}\sum_{m=1}^{M}[g_m(\widehat{z_{r,k+1/2}}) - g_m(z^m_{r,k+1/2})], \widehat{z_{r,k+1/2}} - z\rangle.$$

$A4$ can be bounded with Cauchy-Schwarz (Lemma 5) inequality and Young's inequality (Lemma 6).

$$A4 \leq -\frac{1}{2}\|\widehat{z_{r,k+1/2}} - \widehat{z_{r,k+1}}\|^2 + \eta^c\|\overline{g_{r,k+1/2}} - \overline{g_{r,k}}\|_*\|\widehat{z_{r,k+1/2}} - \widehat{z_{r,k+1}}\|$$
$$\leq -\frac{1}{2}\|\widehat{z_{r,k+1/2}} - \widehat{z_{r,k+1}}\|^2 + \frac{(\eta^c)^2}{2}\|\overline{g_{r,k+1/2}} - \overline{g_{r,k}}\|_*^2 + \frac{1}{2}\|\widehat{z_{r,k+1/2}} - \widehat{z_{r,k+1}}\|^2$$
$$= \frac{(\eta^c)^2}{2}\|\overline{g_{r,k+1/2}} - \overline{g_{r,k}}\|_*^2.$$

Then we have

$$\eta^c\big(\phi(\widehat{z_{r,k+1/2}}) - \phi(z)\big) \leq \tilde{V}^{\ell_{r,k}}_{\omega_{r,k}}(z) - \tilde{V}^{\ell_{r,k+1}}_{\omega_{r,k+1}}(z) - \frac{1}{2}\|\widehat{z_{r,k}} - \widehat{z_{r,k+1/2}}\|^2 + \frac{(\eta^c)^2}{2}\|\overline{g_{r,k+1/2}} - \overline{g_{r,k}}\|_*^2$$
$$+ \eta^c\langle \frac{1}{M}\sum_{m=1}^{M} g_m(z^m_{r,k+1/2}) - \overline{g_{r,k+1/2}}, \widehat{z_{r,k+1/2}} - z\rangle$$
$$+ \eta^c\langle \frac{1}{M}\sum_{m=1}^{M}[g_m(\widehat{z_{r,k+1/2}}) - g_m(z^m_{r,k+1/2})], \widehat{z_{r,k+1/2}} - z\rangle.$$

Taking expectations on both sides we get

$$\eta^c\mathbb{E}\big[\phi(\widehat{z_{r,k+1/2}}) - \phi(z)\big] \leq \tilde{V}^{\ell_{r,k}}_{\omega_{r,k}}(z) - \tilde{V}^{\ell_{r,k+1}}_{\omega_{r,k+1}}(z)$$
$$\underbrace{-\frac{1}{2}\mathbb{E}\big[\|\widehat{z_{r,k}} - \widehat{z_{r,k+1/2}}\|^2\big]}_{B1} + \underbrace{\frac{(\eta^c)^2}{2}\mathbb{E}\big[\|\overline{g_{r,k+1/2}} - \overline{g_{r,k}}\|_*^2\big]}_{B2}$$
$$+ \underbrace{\eta^c\mathbb{E}\big[\langle \frac{1}{M}\sum_{m=1}^{M} g_m(z^m_{r,k+1/2}) - \overline{g_{r,k+1/2}}, \widehat{z_{r,k+1/2}} - z\rangle\big]}_{B3}$$
$$+ \underbrace{\eta^c\mathbb{E}\big[\langle \frac{1}{M}\sum_{m=1}^{M}[g_m(\widehat{z_{r,k+1/2}}) - g_m(z^m_{r,k+1/2})], \widehat{z_{r,k+1/2}} - z\rangle\big]}_{B4}.$$

B2 is bounded in Lemma 14. Therefore, we have

$$
\begin{aligned}
B1 + B2 &\leq \frac{(\eta^c)^2}{2}\Big(\frac{10\sigma^2}{M} + 40\beta^2(\eta^c)^2(k+1)^2 G^2\Big) \\
&\quad + \frac{5(\eta^c)^2\beta^2}{2}\mathbb{E}\Big[\|\widehat{z_{r,k+1/2}} - \widehat{z_{r,k}}\|^2\Big] - \frac{1}{2}\mathbb{E}\Big[\|\widehat{z_{r,k}} - \widehat{z_{r,k+1/2}}\|^2\Big] \\
&= \frac{(\eta^c)^2}{2}\Big(\frac{10\sigma^2}{M} + 40\beta^2(\eta^c)^2(k+1)^2 G^2\Big) + \frac{5(\eta^c)^2\beta^2 - 1}{2}\mathbb{E}\Big[\|\widehat{z_{r,k+1/2}} - \widehat{z_{r,k}}\|^2\Big] \\
&\leq \frac{5\sigma^2(\eta^c)^2}{M} + 20\beta^2(\eta^c)^4(k+1)^2 G^2,
\end{aligned}
$$

for $\eta^c \leq \frac{1}{5^{\frac{1}{2}}\beta}$.

B3 is zero after taking the expectation as shown in Lemma 11. B4 is bounded in Lemma 13. Plugging the bounds for $B1 + B2$, $B3$, and $B4$ back in completes the proof. □

### E.2 HELPING LEMMAS

In this section, we list the helping lemmas that were referenced in the proof of Lemma 1, 2, and 3.

**Lemma 11** (Unbiased Gradient Estimate). *Under Assumption 1 and 2,*

$$
\eta^c \mathbb{E}_{\mathcal{F}_{r,k+1/2}}\Big[\langle \frac{1}{M}\sum_{m=1}^{M} g_m(z_{r,k+1/2}^m) - \overline{g_{r,k+1/2}}, \widehat{z_{r,k+1/2}} - z\rangle\Big] = 0
$$

*Proof.* By the unbiased gradient estimate in Assumption 2b and its following Remark 2,

$$
\begin{aligned}
\eta^c \mathbb{E}_{\mathcal{F}_{r,k+1/2}}&\Big[\langle \frac{1}{M}\sum_{m=1}^{M} g_m(z_{r,k+1/2}^m) - \overline{g_{r,k+1/2}}, \widehat{z_{r,k+1/2}} - z\rangle\Big] \\
&= \eta^c \mathbb{E}_{\mathcal{F}_{r,k}}\Big[\mathbb{E}_{\mathcal{F}_{r,k+1/2}}\big[\langle \frac{1}{M}\sum_{m=1}^{M} g_m(z_{r,k+1/2}^m) - \overline{g_{r,k+1/2}}, \widehat{z_{r,k+1/2}} - z\rangle\big|\mathcal{F}_{r,k}\big]\Big] \\
&= 0.
\end{aligned}
$$

□

**Lemma 12** (Bounded Client Drift under Assumption 2c). $\forall m \in [M]$, $\forall k \in \{0, ..., K-1\}$,
$$
\|\widehat{z_{r,k+1/2}} - z_{r,k+1/2}^m\| \leq 2\eta^c(k+1)G
$$
$$
\|\widehat{z_{r,k}} - z_{r,k}^m\| \leq 2\eta^c k G
$$

*Proof.* By the smoothness of the conjugate of a strongly convex function, i.e., Lemma 9,
$$
\begin{aligned}
\|\widehat{z_{r,k+1/2}} - z_{r,k+1/2}^m\| &= \|\nabla \ell_{r,k}^*(\overline{\omega_{r,k+1/2}}) - \nabla \ell_{r,k}^*(\omega_{r,k+1/2}^m)\| \\
&\leq \|\overline{\omega_{r,k+1/2}} - \omega_{r,k+1/2}^m\|_*
\end{aligned}
$$

After the same round of communication, by the updating sequence, we have $\forall m \in [M]$:
$$
\begin{aligned}
\omega_{r,k+1/2}^m &= \omega_{r,k}^m - \eta^c g_m(z_{r,k}^m; \xi_{r,k}^m) \\
&= -\eta^c \sum_{\ell=0}^{k-1} g_m(z_{r,\ell+1/2}^m; \xi_{r,\ell+1/2}^m) - \eta^c g_m(z_{r,k}^m; \xi_{r,k}^m)
\end{aligned}
$$

Immediately after each round of communication, all machines are synchronized, i.e., $\forall m_1, m_2 \in [M]$, $\omega_{r,0}^{m_1} = \omega_{r,0}^{m_2}$. Therefore, $\forall k \in \{0, ..., K-1\}$,

$$
\begin{aligned}
\omega_{r,k+1/2}^{m_1} - \omega_{r,k+1/2}^{m_2} &= -\eta^c \sum_{\ell=0}^{k-1} g_{m_1}(z_{r,\ell+1/2}^{m_1}; \xi_{r,\ell+1/2}^{m_1}) - \eta^c g_{m_1}(z_{r,k}^{m_1}; \xi_{r,k}^{m_1}) \\
&\quad + \eta^c \sum_{\ell=0}^{k-1} g_{m_2}(z_{r,\ell+1/2}^{m_2}; \xi_{r,\ell+1/2}^{m_2}) + \eta^c g_{m_2}(z_{r,k}^{m_2}; \xi_{r,k}^{m_2})
\end{aligned}
$$

Then $\forall m_1, m_2 \in [M]$, $\forall k \in \{0, ..., K-1\}$, by triangle inequality, Jensen's inequality, and the bounded gradient Assumption 2c,

$$\|\omega_{r,k+1/2}^{m_1} - \omega_{r,k+1/2}^{m_2}\|_* \leq \eta^c \Big( \sum_{\ell=0}^{k-1} \|g_{m_1}(z_{r,\ell+1/2}^{m_1}; \xi_{r,\ell+1/2}^{m_1})\|_* + \|g_{m_1}(z_{r,k}^{m_1}; \xi_{r,k}^{m_1})\|_*$$
$$+ \sum_{\ell=0}^{k-1} \|g_{m_2}(z_{r,\ell+1/2}^{m_2}; \xi_{r,\ell+1/2}^{m_2})\|_* + \|g_{m_2}(z_{r,k}^{m_2}; \xi_{r,k}^{m_2})\|_* \Big)$$
$$\leq 2\eta^c (k+1) G.$$

As a result,

$$\|\widehat{z_{r,k+1/2}} - z_{r,k+1/2}^m\| \leq \|\overline{\omega_{r,k+1/2}} - \omega_{r,k+1/2}^m\|_*$$
$$\leq \sup_{m_1, m_2} \|\omega_{r,k+1/2}^{m_1} - \omega_{r,k+1/2}^{m_2}\|_*$$
$$\leq 2\eta^c (k+1) G.$$

Similarly, we can show that

$$\|\widehat{z_{r,k}} - z_{r,k}^m\| \leq 2\eta^c k G.$$

$\square$

**Lemma 13.** *Under Assumption 1-4,*

$$\eta^c \mathbb{E}\Big[\langle \frac{1}{M} \sum_{m=1}^M [g_m(\widehat{z_{r,k+1/2}}) - g_m(z_{r,k+1/2}^m)], \widehat{z_{r,k+1/2}} - z \rangle\Big] \leq 2^{\frac{3}{2}} \beta (\eta^c)^2 (k+1) G B^{\frac{1}{2}}.$$

*Proof.* The proof of this lemma relies on the bounded client drift in Lemma 12. We start by splitting the inner product using Cauchy-Schwarz inequality in Lemma 5, and state the reference for the following derivation in the parenthesis.

$$\eta^c \mathbb{E}\Big[\langle \frac{1}{M} \sum_{m=1}^M [g_m(\widehat{z_{r,k+1/2}}) - g_m(z_{r,k+1/2}^m)], \widehat{z_{r,k+1/2}} - z \rangle\Big]$$

$$\leq \eta^c \mathbb{E}\Big[\|\frac{1}{M} \sum_{m=1}^M [g_m(\widehat{z_{r,k+1/2}}) - g_m(z_{r,k+1/2}^m)]\|_* \|\widehat{z_{r,k+1/2}} - z\|\Big]$$

$$\leq \eta^c \mathbb{E}\Big[\frac{1}{M} \sum_{m=1}^M \|g_m(\widehat{z_{r,k+1/2}}) - g_m(z_{r,k+1/2}^m)\|_* \|\widehat{z_{r,k+1/2}} - z\|\Big] \qquad \text{(Jensen's)}$$

$$\leq \eta^c \mathbb{E}\Big[\frac{1}{M} \sum_{m=1}^M \beta \|\widehat{z_{r,k+1/2}} - z_{r,k+1/2}^m\|_* \|\widehat{z_{r,k+1/2}} - z\|\Big] \qquad \text{(Smoothness)}$$

$$\leq \eta^c \mathbb{E}\Big[\frac{1}{M} \sum_{m=1}^M 2\beta \eta^c (k+1) G \|\widehat{z_{r,k+1/2}} - z\|\Big] \qquad \text{(Lemma 12)}$$

$$\leq \eta^c \mathbb{E}\Big[2\beta \eta^c (k+1) G \cdot \sqrt{2 V_z^\ell(\widehat{z_{r,k+1/2}})}\Big] \qquad \text{(Strong-convexity of } \ell\text{)}$$

$$\leq 2^{\frac{3}{2}} \beta (\eta^c)^2 (k+1) G B^{\frac{1}{2}} \qquad \text{(Assumption 4)}$$

$\square$

**Lemma 14** (Difference of Gradient and Extra-gradient). *Under Assumption 1-4,*

$$\mathbb{E}\Big[\|\overline{g_{r,k+1/2}} - \overline{g_{r,k}}\|_*^2\Big] \leq \frac{10\sigma^2}{M} + 40\beta^2 (\eta^c)^2 (k+1)^2 G^2 + 5\beta^2 \mathbb{E}\Big[\|\widehat{z_{r,k+1/2}} - \widehat{z_{r,k}}\|^2\Big].$$

*Proof.* By Lemma 7,

$$\mathbb{E}_{\mathcal{F}_{r,k+1/2}}\big[\|\overline{g_{r,k+1/2}} - \overline{g_{r,k}}\|_*^2\big]$$

$$= \mathbb{E}\Big[\big\|\big[\overline{g_{r,k+1/2}} - \frac{1}{M}\sum_{m=1}^M g_m(z_{r,k+1/2}^m)\big]$$

$$+ \big[\frac{1}{M}\sum_{m=1}^M g_m(z_{r,k}^m) - \overline{g_{r,k}}\big] + \frac{1}{M}\sum_{m=1}^M \big[g_m(z_{r,k+1/2}^m) - g_m(\widehat{z_{r,k+1/2}})\big]$$

$$+ \frac{1}{M}\sum_{m=1}^M \big[g_m(\widehat{z_{r,k}}) - g_m(z_{r,k}^m)\big] + \frac{1}{M}\sum_{m=1}^M \big[g_m(\widehat{z_{r,k+1/2}}) - g_m(\widehat{z_{r,k}})\big]\big\|_*^2\Big]$$

$$\leq 5\,\mathbb{E}\Big[\|\overline{g_{r,k+1/2}} - \frac{1}{M}\sum_{m=1}^M g_m(z_{r,k+1/2}^m)\|_*^2\Big] + \underbrace{5\,\mathbb{E}\Big[\|\frac{1}{M}\sum_{m=1}^M g_m(z_{r,k}^m) - \overline{g_{r,k}}\|_*^2\Big]}_{C2}$$

$$+ \underbrace{5\,\mathbb{E}\Big[\|\frac{1}{M}\sum_{m=1}^M \big[g_m(z_{r,k+1/2}^m) - g_m(\widehat{z_{r,k+1/2}})\big]\|_*^2\Big]}_{C3}$$

$$+ \underbrace{5\,\mathbb{E}\Big[\|\frac{1}{M}\sum_{m=1}^M \big[g_m(\widehat{z_{r,k}}) - g_m(z_{r,k}^m)\big]\|_*^2\Big]}_{C4} + \underbrace{5\,\mathbb{E}\Big[\|\frac{1}{M}\sum_{m=1}^M \big[g_m(\widehat{z_{r,k+1/2}}) - g_m(\widehat{z_{r,k}})\big]\|_*^2\Big]}_{C5}$$

For $C1$, by Assumption 2b and its following Remark 2,

$$C1 = \mathbb{E}_{\mathcal{F}_{r,k+1/2}}\Big[\|\frac{1}{M}\sum_{m=1}^M g_m(z_{r,k+1/2}^m; \xi_{r,k+1/2}^m) - \frac{1}{M}\sum_{m=1}^M g_m(z_{r,k+1/2}^m)\|_*^2\Big]$$

$$= \frac{1}{M^2}\mathbb{E}_{\mathcal{F}_{r,k+1/2}}\Big[\|\sum_{m=1}^M \big[g_m(z_{r,k+1/2}^m; \xi_{r,k+1/2}^m) - g_m(z_{r,k+1/2}^m)\big]\|_*^2\Big]$$

$$= \frac{1}{M^2}\operatorname{Var}_{\mathcal{F}_{r,k+1/2}}\Big[\sum_{m=1}^M \big[g_m(z_{r,k+1/2}^m; \xi_{r,k+1/2}^m) - g_m(z_{r,k+1/2}^m)\big]\Big]$$

$$= \frac{1}{M^2}\sum_{m=1}^M \operatorname{Var}_{\mathcal{F}_{r,k+1/2}}\Big[\big[g_m(z_{r,k+1/2}^m; \xi_{r,k+1/2}^m) - g_m(z_{r,k+1/2}^m)\big]\Big] \qquad \text{(i.i.d.)}$$

$$= \frac{1}{M^2}\sum_{m=1}^M \mathbb{E}_{\mathcal{F}_{r,k+1/2}}\Big[\|g_m(z_{r,k+1/2}^m; \xi_{r,k+1/2}^m) - g_m(z_{r,k+1/2}^m)\|_*^2\Big]$$

$$= \frac{1}{M^2}\sum_{m=1}^M \mathbb{E}_{\mathcal{F}_{r,k}}\Big[\mathbb{E}_{\mathcal{F}_{r,k+1/2}}\big[\|g_m(z_{r,k+1/2}^m; \xi_{r,k+1/2}^m) - g_m(z_{r,k+1/2}^m)\|_*^2 | \mathcal{F}_{r,k}\big]\Big]$$

$$\leq \frac{\sigma^2}{M}$$

Similarly, we have $C2 \leq \frac{\sigma^2}{M}$.

For $C3$, by Lemma 7, $\beta$-smoothness of $f_m$, and finally Lemma 12, we have

$$C3 \leq \mathbb{E}\Big[\frac{1}{M^2} \cdot M \sum_{m=1}^M \|g_m(z_{r,k+1/2}^m) - g_m(\widehat{z_{r,k+1/2}})\|_*^2\Big]$$

$$\leq \frac{\beta^2}{M}\sum_{m=1}^M \mathbb{E}\Big[\|z_{r,k+1/2}^m - \widehat{z_{r,k+1/2}}\|^2\Big]$$

$$\leq 4\beta^2(\eta^c)^2(k+1)^2 G^2$$

Similarly for $C4$, we have $C4 \leq 4\beta^2(\eta^c)^2 k^2 G^2$.

For $C5$, by Lemma 7, $\beta$-smoothness of $f_m$ from Assumption 2a, and finally Lemma 12,

$$
\begin{aligned}
C5 &= \mathbb{E}\Big[\frac{1}{M^2}\|\sum_{m=1}^{M}\big[g_m(\widehat{z_{r,k+1/2}}) - g_m(\widehat{z_{r,k}})\big]\|_*^2\Big] \\
&\leq \mathbb{E}\Big[\frac{1}{M^2} \cdot M \sum_{m=1}^{M}\|g_m(\widehat{z_{r,k+1/2}})) - g_m(\widehat{z_{r,k}})\|_*^2\Big] \\
&\leq \beta^2 \mathbb{E}\Big[\|\widehat{z_{r,k+1/2}} - \widehat{z_{r,k}}\|^2\Big].
\end{aligned}
$$

Plugging the bounds for $C1,\ C2, C3, C4,$ and $C5$ back in completes the proof. $\qquad\square$

## F  COMPLETE ANALYSIS OF FEDUALEX FOR COMPOSITE CONVEX OPTIMIZATION

In this section, we reduce the problem to composite convex optimization in the following form:

$$\min_{x\in\mathcal{X}} \phi(x) = f(x) + \psi(x) \tag{9}$$

where $f(x) = \frac{1}{M}\sum_{m=1}^{M} f_m(x)$. The analysis builds upon the strong-convexity of the distance-generating function $h$ in Assumption 3 and the following set of assumptions in the convex optimization setting:

**Assumption 5.** *We make the following assumptions:*

a. *(Convexity of $f$)* $\forall m \in [M]$, $f_m$ *is convex. That is,* $\forall x, x' \in \mathcal{X}$,

$$f_m(x) - f_m(x') \leq \langle f_m(x), x - x'\rangle.$$

b. *(Local Smoothness of $f$)* $\forall m \in [M]$, $f_m$ *is $\beta$-smooth:* $\forall x, x' \in \mathcal{X}$,

$$f_m(x) \leq f_m(x') + \langle f_m(x'), x - x'\rangle + \frac{\beta}{2}\|x - x'\|.$$

c. *(Convexity of $\psi$)* $\psi(x)$ *is convex.*

d. *(Local Unbiased Estimate and Bounded Variance) For any client $m \in [M]$, the local gradient queried by some local random sample $\xi^m$ is unbiased and also bounded in variance, i.e.,* $\mathbb{E}_\xi[g_m(x^m;\xi^m)] = g_m(x^m)$ *and* $\mathbb{E}_\xi[\|g_m(x_m;\xi_m) - g_m(x_m)\|_*^2] \leq \sigma^2$.

e. *(Bounded Gradient)* $\forall m \in [M]$, $\|g_m(x_m;\xi_m)\|_* \leq G$.

Federated dual extrapolation for composite convex optimization is to replace the part of Algorithm 1 highlighted in green with the following updating sequence, where we overuse $\varsigma$ now as the notation for dual variables in the convex setting as well.

$$
\begin{aligned}
&\varsigma_{r,0}^m = \varsigma_r \\
&\textbf{for } k = 0, 1, \ldots, K-1 \textbf{ do} \\
&\qquad x_{r,k}^m = \widetilde{\text{Prox}}_{\bar{\varsigma}}^{h_{r,k}}(\varsigma_{r,k}^m) \\
&\qquad x_{r,k+1/2}^m = \widetilde{\text{Prox}}_{\bar{\varsigma}-\varsigma_{r,k}^m}^{h_{r,k+1}}(\eta^c g_m(x_{r,k}^m; \xi_{r,k}^m)) \\
&\qquad \varsigma_{r,k+1}^m = \varsigma_{r,k}^m + \eta^c g_m(x_{r,k+1/2}^m; \xi_{r,k+1/2}^m) \\
&\textbf{end for}
\end{aligned}
$$

For the proximal operator defined by $h_{r,k}$, reformulating from its Definition 4 to $\nabla h_{r,k}^*$ in Definition 10 yields

$$x_{r,k}^m = \arg\min_x\{\langle\varsigma_{r,k}^m - \bar{\varsigma}, x\rangle + h_{r,k}(x)\} = \nabla h_{r,k}^*(\bar{\varsigma} - \varsigma_{r,k}^m)$$

$$x_{r,k+1/2}^m = \arg\min_x\{\langle\eta^c g_m(x_{r,k}^m;\xi_{r,k}^m) - (\bar{\varsigma} - \varsigma_{r,k}^m), x\rangle + h_{r,k+1}(x)\} = \nabla h_{r,k+1}^*((\bar{\varsigma} - \varsigma_{r,k}^m) - \eta^c g_m(x_{r,k}^m;\xi_{r,k}^m))$$

$$\varsigma_{r,k+1}^m = \varsigma_{r,k}^m + \eta^c g_m(x_{r,k+1/2}^m;\xi_{r,k+1/2}^m)$$

Similarly, we define auxiliary dual variable $\mu_{r,k}^m = \bar{\varsigma} - \varsigma_{r,k}^m$ and $\mu_{r,k+1/2}^m$ the dual image of $x_{r,k+1/2}^m$. Then by definition, $x_{r,k}^m = \nabla h_{r,k}^*(\mu_{r,k}^m)$ and $x_{r,k+1/2}^m = \nabla h_{r,k+1}^*(\mu_{r,k+1/2}^m)$. The updating sequence is equivalent to $\mu_{r,k+1/2}^m = \mu_{r,k}^m - \eta g_m(x_{r,k}^m; \xi_{r,k}^m)$ followed by $\mu_{r,k+1}^m = \mu_{r,k}^m - \eta g_m(x_{r,k+1/2}^m; \xi_{r,k+1/2}^m)$. For the shadow sequence of averaged variables $\overline{\mu_{r,k}} = \frac{1}{M}\sum_{m=1}^M \mu_{r,k}^m$ and $\overline{g_{r,k}} = \frac{1}{M}\sum_{m=1}^M g_m(x_{r,k}^m; \xi_{r,k}^m)$,

$$\overline{\mu_{r,k+1/2}} = \overline{\mu_{r,k}} - \eta^c \overline{g_{r,k}}, \tag{10}$$

$$\overline{\mu_{r,k+1}} = \overline{\mu_{r,k}} - \eta^c \overline{g_{r,k+1/2}}. \tag{11}$$

Finally, the projections of the averaged dual back to the primal space are $\widehat{x_{r,k}} = \nabla h_{r,k}^*(\overline{\mu_{r,k}})$ and $\widehat{x_{r,k+1/2}} = \nabla h_{r,k+1}^*(\overline{\mu_{r,k+1/2}})$

**Theorem 2.** *Under Assumption 5, the ergodic intermediate sequence generated by FeDualEx for composite convex objectives satisfies*

$$\mathbb{E}\big[\phi(\frac{1}{RK}\sum_{r=0}^{R-1}\sum_{k=0}^{K-1}\widehat{x_{r,k+1/2}}) - \phi(x)\big] \leq \frac{B}{\eta^c RK} + 20\beta^2(\eta^c)^3 K^2 G^2 + \frac{5\sigma^2\eta^c}{M} + 2\beta(\eta^c)^3 K^2 G^2.$$

*Choosing step size*

$$\eta^c = \min\{\frac{1}{5^{\frac{1}{2}}\beta}, \frac{B^{\frac{1}{4}}}{20^{\frac{1}{4}}\beta^{\frac{1}{2}}G^{\frac{1}{2}}K^{\frac{3}{4}}R^{\frac{1}{4}}}, \frac{B^{\frac{1}{2}}M^{\frac{1}{2}}}{5^{\frac{1}{2}}\sigma R^{\frac{1}{2}}K^{\frac{1}{2}}}, \frac{B^{\frac{1}{3}}}{2^{\frac{1}{3}}\beta^{\frac{1}{3}}G^{\frac{2}{3}}KR^{\frac{1}{3}}}\}$$

*further yields the following convergence rate:*

$$\mathbb{E}\big[\phi(\frac{1}{RK}\sum_{r=0}^{R-1}\sum_{k=0}^{K-1}\widehat{x_{r,k+1/2}}) - \phi(x)\big] \leq \frac{5^{\frac{1}{2}}\beta B}{RK} + \frac{20^{\frac{1}{4}}\beta^{\frac{1}{2}}G^{\frac{1}{2}}B^{\frac{3}{4}}}{K^{\frac{1}{4}}R^{\frac{3}{4}}} + \frac{5^{\frac{1}{2}}\sigma B^{\frac{1}{2}}}{M^{\frac{1}{2}}R^{\frac{1}{2}}K^{\frac{1}{2}}} + \frac{2^{\frac{1}{3}}\beta^{\frac{1}{3}}G^{\frac{2}{3}}B^{\frac{2}{3}}}{R^{\frac{2}{3}}}.$$

*Proof.* As the proof for Theorem 1, the proof for this theorem depends on Lemma 15 and Lemma 16, which further yield Lemma 17. These lemmas are presented and proved right after this theorem. Here, we start from Lemma 17. Telescoping over all $k \in \{0, ..., K-1\}$ and all $r \in \{0, ..., R-1\}$ assuming $\eta^s = 1$ yields

$$\eta^c \mathbb{E}\big[\sum_{r=0}^{R-1}\sum_{k=0}^{K-1}\phi(\widehat{x_{r,k+1/2}}) - RK\phi(x)\big] \leq \tilde{V}_{\overline{\mu_{0,0}}}^{h_{0,0}}(x) - \tilde{V}_{\overline{\mu_{R,K}}}^{h_{R,K}}(x) + \frac{5\sigma^2(\eta^c)^2 KR}{M}$$

$$+ 20\beta^2(\eta^c)^4 K^3 RG^2 + 2\beta(\eta^c)^3 K^3 RG^2.$$

By Assumption 4, $\tilde{V}_{\overline{\mu_{0,0}}}^{h_{0,0}}(x) = V_{x_0}^h(x) \leq B$, where $x_0 = \nabla h^*(\bar{\varsigma})$. Dividing both sides by $\eta^c KR$ followed by applying Jensen's inequality (Lemma 4) completes the proof. $\square$

**Lemma 15** (Bounding the Regularization Term). $\forall x,$

$$\eta^c\big[\psi(\widehat{x_{r,k+1/2}}) - \psi(x)\big] = \tilde{V}_{\overline{\mu_{r,k}}}^{h_{r,k}}(x) - \tilde{V}_{\overline{\mu_{r,k+1}}}^{h_{r,k+1}}(x) - \tilde{V}_{\overline{\mu_{r,k}}}^{h_{r,k}}(\widehat{x_{r,k+1/2}}) - \tilde{V}_{\overline{\mu_{r,k+1/2}}}^{h_{r,k+1}}(\widehat{x_{r,k+1}})$$

$$+ \eta^c\langle\overline{g_{r,k+1/2}} - \overline{g_{r,k}}, \widehat{x_{r,k+1/2}} - \widehat{x_{r,k+1}}\rangle + \eta^c\langle\overline{g_{r,k+1/2}}, x - \widehat{x_{r,k+1/2}}\rangle$$

*Proof.* The proof of this Lemma is almost identical to the proof of Lemma 1 with a mere change of variables and distance-generating function from saddle point setting to convex setting. $\square$

The following Lemma highlights the primary difference in the analysis of convex optimization and saddle point optimization. The smoothness of $f_m$ provides an alternative presentation to gradient Lipschitzness that establishes the connection between $\widehat{x_{r,k+1/2}}$, the primal projection of averaged dual on the central server, and $x_{r,k+1/2}^m$ on each client.

**Lemma 16** (Bounding the Smooth Term). $\forall x,$

$$f(\widehat{x_{r,k+1/2}}) - f(x) \leq \langle\overline{g_{r,k+1/2}}, \widehat{x_{r,k+1/2}} - x\rangle + \langle\frac{1}{M}\sum_{m=1}^M g_m(x_{r,k+1/2}^m) - \overline{g_{r,k+1/2}}, \widehat{x_{r,k+1/2}} - x\rangle$$

$$+ \frac{\beta}{2M}\sum_{m=1}^M\|\widehat{x_{r,k+1/2}} - x_{r,k+1/2}^m\|^2.$$

*Proof.* By the smoothness $f_m$ in the form of Assumption 5b and then the convexity of $f_m$ in the form of Assumption 5a,

$$f_m(\widehat{x_{r,k+1/2}}) \leq f_m(x_{r,k+1/2}^m) + \langle g_m(x_{r,k+1/2}^m), \widehat{x_{r,k+1/2}} - x_{r,k+1/2}^m \rangle + \frac{\beta}{2} \|\widehat{x_{r,k+1/2}} - x_{r,k+1/2}^m\|^2$$

$$\leq f_m(x_{r,k+1/2}^m) + \langle g_m(x_{r,k+1/2}^m), \widehat{x_{r,k+1/2}} - x_{r,k+1/2}^m \rangle + \frac{\beta}{2} \|\widehat{x_{r,k+1/2}} - x_{r,k+1/2}^m\|^2$$

$$+ f_m(x) - f_m(x_{r,k+1/2}^m) + \langle g_m(x_{r,k+1/2}^m), x_{r,k+1/2}^m - x \rangle$$

$$\leq f_m(x) + \langle g_m(x_{r,k+1/2}^m), \widehat{x_{r,k+1/2}} - x \rangle + \frac{\beta}{2} \|\widehat{x_{r,k+1/2}} - x_{r,k+1/2}^m\|^2$$

Then for function $f = \frac{1}{M} \sum_{m=1}^M f_m$,

$$f(\widehat{x_{r,k+1/2}}) - f(x) \leq \frac{1}{M} \sum_{m=1}^M \left[ f_m(\widehat{x_{r,k+1/2}}) - f_m(x) \right]$$

$$\leq \langle \frac{1}{M} \sum_{m=1}^M g_m(x_{r,k+1/2}^m), \widehat{x_{r,k+1/2}} - x \rangle + \frac{1}{M} \sum_{m=1}^M \frac{\beta}{2} \|\widehat{x_{r,k+1/2}} - x_{r,k+1/2}^m\|^2$$

$$= \langle \overline{g_{r,k+1/2}}, \widehat{x_{r,k+1/2}} - x \rangle + \langle \frac{1}{M} \sum_{m=1}^M g_m(x_{r,k+1/2}^m) - \overline{g_{r,k+1/2}}, \widehat{x_{r,k+1/2}} - x \rangle$$

$$+ \frac{\beta}{2M} \sum_{m=1}^M \|\widehat{x_{r,k+1/2}} - x_{r,k+1/2}^m\|^2.$$

$\square$

Now we are ready to present the main lemma that combines Lemma 15 and Lemma 16. For the proof, we utilize again Lemma 11, Lemma 12, and Lemma 14, all of which we claim to hold trivially in the composite convex optimization setting.

**Lemma 17** (Main Lemma for FeDualEx in Composite Convex Optimization). *Under Assumption 5,*

$$\eta^c \mathbb{E}\left[\phi(\widehat{x_{r,k+1/2}}) - \phi(x)\right] \leq \tilde{V}_{\mu_{r,k}}^{h_{r,k}}(x) - \tilde{V}_{\mu_{r,k+1}}^{h_{r,k+1}}(x) + \frac{5\sigma^2 \eta^c}{M} + 10\beta^2(\eta^c)^3(2k^2 + 2k + 1)G^2$$

$$+ \frac{(\eta^c)^2 \sigma^2}{2M(1 - \eta^c)} + 2\beta(\eta^c)^3(k+1)^2 G^2.$$

*Proof.* Summing the results in Lemma 15 and Lemma 16:

$$\eta^c\left(\phi(\widehat{x_{r,k+1/2}}) - \phi(x)\right) \leq \tilde{V}_{\mu_{r,k}}^{h_{r,k}}(x) - \tilde{V}_{\mu_{r,k+1}}^{h_{r,k+1}}(x) - \tilde{V}_{\mu_{r,k}}^{h_{r,k}}(\widehat{x_{r,k+1/2}}) - \tilde{V}_{\mu_{r,k+1/2}}^{h_{r,k+1}}(\widehat{x_{r,k+1}})$$

$$+ \eta^c \langle \overline{g_{r,k+1/2}} - \overline{g_{r,k}}, \widehat{x_{r,k+1/2}} - \widehat{x_{r,k+1}} \rangle + \frac{\eta^c \beta}{2M} \sum_{m=1}^M \|\widehat{x_{r,k+1/2}} - x_{r,k+1/2}^m\|^2$$

$$+ \eta^c \langle \frac{1}{M} \sum_{m=1}^M g_m(x_{r,k+1/2}^m) - \overline{g_{r,k+1/2}}, \widehat{x_{r,k+1/2}} - x \rangle.$$

For the latter two generalized Bregman divergence terms $-\tilde{V}_{\mu_{r,k}}^{h_{r,k}}(\widehat{x_{r,k+1/2}}) - \tilde{V}_{\mu_{r,k+1/2}}^{h_{r,k+1}}(\widehat{x_{r,k+1}})$, we bound them by Lemma 10 and the strong convexity of $h$ in Assumption 3. As a result,

$$
\begin{aligned}
\eta^c\big(\phi(\widehat{x_{r,k+1/2}}) - \phi(x)\big) \leq{}& \tilde{V}_{\mu_{r,k}}^{h_{r,k}}(x) - \tilde{V}_{\mu_{r,k+1}}^{h_{r,k+1}}(x) - \frac{1}{2}\|\widehat{x_{r,k}} - \widehat{x_{r,k+1/2}}\|^2 \\
& \underbrace{- \frac{1}{2}\|\widehat{x_{r,k+1/2}} - \widehat{x_{r,k+1}}\|^2 + \eta^c\langle \overline{g_{r,k+1/2}} - \overline{g_{r,k}}, \widehat{x_{r,k+1/2}} - \widehat{x_{r,k+1}}\rangle}_{A} \\
& + \langle \frac{\eta^c}{M}\sum_{m=1}^{M} g_m(x_{r,k+1/2}^m) - \overline{g_{r,k+1/2}}, \widehat{x_{r,k+1/2}} - x\rangle \\
& + \frac{\eta^c\beta}{2M}\sum_{m=1}^{M}\|\widehat{x_{r,k+1/2}} - x_{r,k+1/2}^m\|^2.
\end{aligned}
$$

$A$ can be bounded with Cauchy-Schwarz inequality (Lemma 5) and Young's inequality (Lemma 6).

$$
\begin{aligned}
A \leq{}& -\frac{1}{2}\|\widehat{x_{r,k+1/2}} - \widehat{x_{r,k+1}}\|^2 + \eta^c\|\overline{g_{r,k+1/2}} - \overline{g_{r,k}}\|_*\|\widehat{x_{r,k+1/2}} - \widehat{x_{r,k+1}}\| \\
\leq{}& -\frac{1}{2}\|\widehat{x_{r,k+1/2}} - \widehat{x_{r,k+1}}\|^2 + \frac{(\eta^c)^2}{2}\|\overline{g_{r,k+1/2}} - \overline{g_{r,k}}\|_*^2 + \frac{1}{2}\|\widehat{x_{r,k+1/2}} - \widehat{x_{r,k+1}}\|^2 \\
={}& \frac{(\eta^c)^2}{2}\|\overline{g_{r,k+1/2}} - \overline{g_{r,k}}\|_*^2.
\end{aligned}
$$

Taking expectations on both sides we get

$$
\begin{aligned}
\eta^c\mathbb{E}\big[\phi(\widehat{x_{r,k+1/2}}) - \phi(x)\big] \leq{}& \tilde{V}_{\mu_{r,k}}^{h_{r,k}}(x) - \tilde{V}_{\mu_{r,k+1}}^{h_{r,k+1}}(x) \\
& \underbrace{-\frac{1}{2}\mathbb{E}\big[\|\widehat{x_{r,k}} - \widehat{x_{r,k+1/2}}\|^2\big]}_{B1} + \underbrace{\frac{(\eta^c)^2}{2}\mathbb{E}\big[\|\overline{g_{r,k+1/2}} - \overline{g_{r,k}}\|_*^2\big]}_{B2} \\
& + \underbrace{\mathbb{E}\big[\langle \frac{\eta^c}{M}\sum_{m=1}^{M} g_m(x_{r,k+1/2}^m) - \overline{g_{r,k+1/2}}, \widehat{x_{r,k+1/2}} - x\rangle\big]}_{B3} \\
& + \underbrace{\frac{\eta^c\beta}{2M}\sum_{m=1}^{M}\mathbb{E}\big[\|\widehat{x_{r,k+1/2}} - x_{r,k+1/2}^m\|^2\big]}_{B4}.
\end{aligned}
$$

B2 is bounded in Lemma 14. Therefore, for $\eta^c \leq \frac{1}{5^{\frac{1}{2}}\beta}$,

$$
B1 + B2 \leq \frac{5\sigma^2(\eta^c)^2}{M} + 20\beta^2(\eta^c)^4(k+1)^2G^2.
$$

B3 is zero after taking the expectation by Lemma 11. B4 is bounded in Lemma 12. Plugging the bounds for $B1 + B2$, $B3$, and $B4$ back in completes the proof. $\qquad\square$

## G   FeDualEx in Other Settings

In this section, we provide the algorithm along with the convergence rate for sequential versions of FeDualEx. The proofs in this section rely only on the Lipschitzness of the gradient operator. As a result, the analysis applies to both composite saddle point optimization and composite convex optimization.

### G.1   Stochastic Dual Extrapolation for Composite Saddle Point Optimization

The sequential version of FeDualEx immediately yields Algorithm 3, stochastic dual extrapolation for Composite SPP. This algorithm generalizes dual extrapolation to both composite and smooth

---

**Algorithm 3** STOCHASTIC-DUAL-EXTRAPOLATION for Composite SPP

---

**Input:** $\phi(z) = f(x, y) + \psi_1(x) - \psi_2(y)$: objective function; $\ell(z)$: distance-generating function;
$\quad g(z) = (\nabla_x f(x, y), -\nabla_y f(x, y))$: gradient operator.
**Hyperparameters:** $T$: number of iterations; $\eta$: step size.
**Dual Initialization:** $\varsigma_0 = 0$: initial dual variable, $\bar{\varsigma} \in \mathcal{S}$: fixed point in the dual space.
**Output:** Approximate solution $z = (x, y)$ to $\min_{x \in \mathcal{X}} \max_{y \in \mathcal{Y}} \phi(x, y)$

$\quad$ **for** $t = 0, 1, \ldots, T-1$ **do**
$\qquad z_t = \tilde{\text{Prox}}^{\ell_t}_{\bar{\varsigma}}(\varsigma_t)$ $\qquad\qquad\qquad\qquad\qquad$ ▷ Two-step evaluation of the generalized proximal operator
$\qquad z_{t+1/2} = \tilde{\text{Prox}}^{\ell_t}_{\bar{\varsigma}-\varsigma_t}(\eta^c g(z_t; \xi_t))$
$\qquad \varsigma_{t+1} = \varsigma_t + \eta^c g(z_{t+1/2}; \xi_{t+1/2})$ $\qquad\qquad\qquad\qquad\qquad$ ▷ Dual variable update
$\quad$ **end for**
$\quad$ **Return:** $\frac{1}{T} \sum_{t=0}^{T-1} z_{t+1/2}$.

---

stochastic saddle point optimization with the latter taking $\psi(z) = 0$. Its convergence rate is analyzed in the following theorem, which to the best of our knowledge, is the first one for stochastic composite saddle point optimization.

**Theorem 3.** *Under the sequential version of Assumption 1-4, namely with $M = 1$, $\forall z \in \mathcal{Z}$, the ergodic intermediate sequence generated by Algorithm 3 satisfies*

$$\mathbb{E}\big[\text{Gap}(\frac{1}{T}\sum_{t=0}^{T-1} z_{t+1/2})\big] \leq \frac{B}{\eta T} + 3\sigma^2 \eta.$$

*Choosing step size*

$$\eta = \min\{\frac{1}{3^{\frac{1}{2}}\beta}, \frac{B^{\frac{1}{2}}}{3^{\frac{1}{2}}\sigma T^{\frac{1}{2}}}\},$$

*further yields the following convergence rate:*

$$\mathbb{E}\big[\text{Gap}(\frac{1}{T}\sum_{t=0}^{T-1} z_{t+1/2})\big] \leq \frac{3^{\frac{1}{2}}\beta B}{T} + \frac{3^{\frac{1}{2}}\sigma B^{\frac{1}{2}}}{T^{\frac{1}{2}}}.$$

*Proof.* By proof similar to Lemma 1, we have

$$\eta\big[\psi(z_{t+1/2}) - \psi(z)\big] = \tilde{V}^{\ell_t}_{\omega_t}(z) - \tilde{V}^{\ell_{t+1}}_{\omega_{t+1}}(z) - \tilde{V}^{\ell_t}_{\omega_t}(z_{t+1/2}) - \tilde{V}^{\ell_{t+1}}_{\omega_{t+1/2}}(z_{t+1})$$
$$+ \eta\langle g_{t+1/2} - g_t, z_{t+1/2} - z_{t+1}\rangle + \eta\langle g_{t+1/2}, z - z_{t+1/2}\rangle$$
$$\leq \tilde{V}^{\ell_t}_{\omega_t}(z) - \tilde{V}^{\ell_{t+1}}_{\omega_{t+1}}(z)$$
$$\underbrace{- \frac{1}{2}\|z_t - z_{t+1/2}\|^2 - \frac{1}{2}\|z_{t+1/2} - z_{t+1}\|^2 + \eta\langle g_{t+1/2} - g_t, z_{t+1/2} - z_{t+1}\rangle}_{A}$$
$$+ \underbrace{\eta\langle g(z_{t+1/2}) - g_{t+1/2}, z_{t+1/2} - z\rangle}_{B} - \eta\langle g(z_{t+1/2}), z_{t+1/2} - z\rangle.$$

where the inequality holds by Lemma 10 and the strong convexity of $\ell$ in Remark 4, and then simply expanding the last term to build a connection between the stochastic gradient and true gradient. By

---

**Algorithm 4** COMPOSITE-DUAL-EXTRAPOLATION

---

**Input:** $\phi(z) = f(x,y) + \psi_1(x) - \psi_2(y)$: objective function; $\ell(z)$: distance-generating function; $g(z) = (\nabla_x f(x,y), -\nabla_y f(x,y))$: gradient operator.

**Hyperparameters:** $T$: number of iterations; $\eta$: step size.

**Dual Initialization:** $\varsigma_0 = 0$: initial dual variable, $\bar{\varsigma} \in \mathcal{S}$: fixed point in the dual space.

**Output:** Approximate solution $z = (x,y)$ to $\min_{x \in \mathcal{X}} \max_{y \in \mathcal{Y}} \phi(x,y)$

    **for** $t = 0, 1, \ldots, T-1$ **do**

      $z_t = \tilde{\text{Prox}}_{\bar{\varsigma}}^{\ell_t}(\varsigma_t)$           $\triangleright$ Two-step evaluation of the generalized proximal operator

      $z_{t+1/2} = \tilde{\text{Prox}}_{\bar{\varsigma}-\varsigma_t}^{\ell_t}(\eta^c g(z_t))$

      $\varsigma_{t+1} = \varsigma_t + \eta^c g(z_{t+1/2})$           $\triangleright$ Dual variable update

    **end for**

    **Return:** $\frac{1}{T} \sum_{t=0}^{T-1} z_{t+1/2}$.

---

Cauchy-Schwarz inequality (Lemma 5), Young's inequality (Lemma 6), and Lemma 7,

$$A \leq -\frac{1}{2}\|z_t - z_{t+1/2}\|^2 - \frac{1}{2}\|z_{t+1/2} - z_{t+1}\|^2 + \frac{\eta^2}{2}\|g_{t+1/2} - g_t\|_*^2 + \frac{1}{2}\|z_{t+1/2} - z_{t+1}\|^2$$

$$= -\frac{1}{2}\|z_t - z_{t+1/2}\|^2 + \frac{\eta^2}{2}\|[g_{t+1/2} - g(z_{t+1/2})] + [g(z_t) - g_t] + [g(z_{t+1/2}) - g(z_t)]\|_*^2$$

$$\leq -\frac{1}{2}\|z_t - z_{t+1/2}\|^2 + \frac{3\eta^2}{2}\|g(z_{t+1/2}) - g(z_t)\|_*^2$$

$$+ \frac{3\eta^2}{2}\|g_{t+1/2} - g(z_{t+1/2})\|_*^2 + \frac{3\eta^2}{2}\|g(z_t) - g_t\|_*^2$$

$$\leq \frac{3\eta^2\beta^2 - 1}{2}\|z_t - z_{t+1/2}\|^2 + \frac{3\eta^2}{2}\|g_{t+1/2} - g(z_{t+1/2})\|_*^2 + \frac{3\eta^2}{2}\|g(z_t) - g_t\|_*^2,$$

where the last inequality holds by the $\beta$-Lipschitzness of the gradient operator. After taking expectations, the last two terms are bounded by the variance of the gradient $\sigma^2$, and $B$ becomes zero by proof similar to Lemma 11. Therefore, for $\eta \leq \frac{1}{\sqrt{3}\beta}$

$$\eta\mathbb{E}\big[\langle g(z_{t+1/2}), z_{t+1/2} - z\rangle + \psi(z_{t+1/2}) - \psi(z)\big] \leq \tilde{V}_{\omega_t}^{\ell_t}(z) - \tilde{V}_{\omega_{t+1}}^{\ell_{t+1}}(z) + 3\eta^2\sigma^2.$$

Telescoping over all $t \in \{0, ..., T-1\}$ and dividing both sides by $\eta T$ completes the proof. $\qquad \square$

### G.2   DETERMINISTIC DUAL EXTRAPOLATION FOR COMPOSITE SADDLE POINT OPTIMIZATION

Further removing the data-dependent noise in the gradient, we present the deterministic sequential version of FeDualEx, which still generalizes Nesterov's dual extrapolation (Nesterov, 2007) to composite saddle point optimization. As a result, we term this algorithm composite dual extrapolation, as presented in Algorithm 4.

We also provide a convergence analysis, which shows that composite dual extrapolation achieves the $\mathcal{O}(\frac{1}{T})$ convergence rate as its original non-composite smooth version (Nesterov, 2007), as well as composite mirror prox (CoMP) (He et al., 2015). We do so with a very simple proof based on the recently proposed notion of relative Lipschitzness (Cohen et al., 2021). We start by introducing the definition of relative Lipschitzness and a relevant lemma.

**Definition 11** (Relative Lipschitzness (Definition 1 in Cohen et al. (2021))). *For convex distance-generating function $h : \mathcal{Z} \to \mathbb{R}$, we call operator $g : \mathcal{Z} \to \mathcal{Z}^*$ $\lambda$-relatively Lipschitz with respect to $h$ if $\forall z, w, u \in \mathcal{Z}$,*

$$\langle g(w) - g(z), w - u\rangle \leq \lambda(V_z^h(w) + V_w^h(u)).$$

**Lemma 18** (Lemma 1 in Cohen et al. (2021)). *If $g$ is $\beta$-Lipschitz and $h$ is $\alpha$-strongly convex, $g$ is $\frac{\beta}{\alpha}$-relatively Lipschitz with respect to $h$.*

**Theorem 4.** *Under the basic convexity assumption and $\beta$-Lipschitzness of $g$, $\forall z \in \mathcal{Z}$ and $\eta \leq \frac{1}{\beta}$, composite dual extrapolation satisfies $\text{Gap}(\frac{1}{T}\sum_{t=0}^{T-1} z_{t+1/2}) \leq \frac{\beta B}{T}$.*

*Proof.* By proof similar to Lemma 1, we have

$$\eta\big[\psi(z_{t+1/2}) - \psi(z)\big] = \tilde{V}_{\omega_t}^{\ell_t}(z) - \tilde{V}_{\omega_{t+1}}^{\ell_{t+1}}(z) - \tilde{V}_{\omega_t}^{\ell_t}(z_{t+1/2}) - \tilde{V}_{\omega_{t+1/2}}^{\ell_{t+1}}(z_{t+1})$$
$$+ \eta\langle g(z_{t+1/2}) - g(z_t), z_{t+1/2} - z_{t+1}\rangle + \eta\langle g(z_{t+1/2}), z - z_{t+1/2}\rangle.$$

By Lemma 18, we know that $g$ is $\beta$-relatively Lipschitz with respect to $\ell$ under the $\beta$-Lipschitzness assumption of $g$ and 1-strong convexity assumption of $\ell$. Then by Definition 11, we have

$$\eta\big[\psi(z_{t+1/2}) - \psi(z) + \langle g(z_{t+1/2}), z_{t+1/2} - z\rangle\big]$$
$$\leq \tilde{V}_{\omega_t}^{\ell_t}(z) - \tilde{V}_{\omega_{t+1}}^{\ell_{t+1}}(z) - \tilde{V}_{\omega_t}^{\ell_t}(z_{t+1/2}) - \tilde{V}_{\omega_{t+1/2}}^{\ell_{t+1}}(z_{t+1}) + \eta^c\langle g(z_{t+1/2}) - g(z_t), z_{t+1/2} - z_{t+1}\rangle$$
$$\leq \tilde{V}_{\omega_t}^{\ell_t}(z) - \tilde{V}_{\omega_{t+1}}^{\ell_{t+1}}(z) - \tilde{V}_{\omega_t}^{\ell_t}(z_{t+1/2}) - \tilde{V}_{\omega_{t+1/2}}^{\ell_{t+1}}(z_{t+1}) + \eta^c\beta\big[V_{z_t}^{\ell}(z_{t+1/2}) + V_{z_{t+1/2}}^{\ell}(z_{t+1})\big]$$
$$\leq \tilde{V}_{\omega_t}^{\ell_t}(z) - \tilde{V}_{\omega_{t+1}}^{\ell_{t+1}}(z).$$

where the last inequality holds for $\eta \leq \frac{1}{\beta}$ by Lemma 10. Telescoping over all $t \in \{0, ..., T-1\}$ and dividing both sides by $\eta T$ completes the proof. $\qquad\square$

## H  FEDERATED MIRROR PROX

We present Federated Mirror Prox (FedMiP) here in Algorithm 2 as a baseline. The part highlighted in green resembles the mirror prox algorithm introduced in Section C.1.2. We use the composite mirror map representation introduced in Section C.1.1 to avoid confusion, as the composite proximal operator we proposed for FeDualEx is slightly different from that used in composite mirror descent as discussed in Section 4.1.

---

**Algorithm 2** FEDERATED-MIRROR-PROX (FedMiP) for Composite SPP

---

**Input:** $\phi(z) = f(x,y) + \psi_1(x) - \psi_2(y) = \frac{1}{M}\sum_{m=1}^{M} f_m(\cdot) + \psi_1(x) - \psi_2(y)$: objective function; $\ell(z)$: distance-generating function; $g_m(z) = (\nabla_x f_m(x,y), -\nabla_y f_m(x,y))$: gradient operator.
**Hyperparameters:** $R$: number of rounds of communication; $K$: number of local update iterations; $\eta^s$: server step size; $\eta^c$: client step size.
**Primal Initialization:** $z_0$: initial primal variable.
**Output:** Approximate solution $z = (x,y)$ to $\min_{x \in \mathcal{X}} \max_{y \in \mathcal{Y}} \phi(x,y)$
1: **for** $r = 0, 1, \ldots, R-1$ **do**
2:     Sample a subset of clients $C_r \subseteq [M]$
3:     **for** $m \in C_r$ **in parallel do**
4:         $z_{r,0}^m = z_r$
5:         **for** $k = 0, 1, \ldots, K-1$ **do**
6:             $z_{r,k+1/2}^m = \nabla(\ell + \eta^c\psi)^*(\nabla h(z_{r,k}^m) - \eta^c g(z_{r,k}^m; \xi_{r,k}^m))$
7:             $z_{r,k+1}^m = \nabla(\ell + \eta^c\psi)^*(\nabla h(z_{r,k}^m) - \eta^c g(z_{r,k+1/2}^m; \xi_{r,k+1/2}^m))$
8:         **end for**
9:     **end parallel for**
10:    $\Delta_r = \frac{1}{|\mathcal{C}_r|}\sum_{m \in \mathcal{C}_r}(z_{r,K}^m - z_{r,0}^m)$
11:    $z_{r+1} = \nabla(\ell + \eta^s\eta^c K\psi)^*(\nabla h(z_r) + \eta^s\Delta_r)$
12: **end for**
13: **Return:** $\frac{1}{RK}\sum_{r=0}^{R-1}\sum_{k=0}^{K-1} z_{r,k+1/2}$.

---

