# OpenReview forum: "Local Composite Saddle Point Optimization"
_ICLR.cc/2024/Conference — ICLR 2024 poster_

### Official Review · Reviewer_geTM · 2023-11-01

**Soundness:** 3 good
**Presentation:** 3 good
**Contribution:** 3 good
**Rating:** 6
**Confidence:** 4

**Summary:**

The paper proposes FeDualEx a federated primal-dual algorithm for solving distributed composite saddle point problems. The authors consider a homogeneous setting and provide convergence guarantees achieved by FeDualEx when the objective function is convex-concave. The authors also evaluate the proposed algorithm experimentally on synthetic and real datasets.

**Strengths:**

Overall. the paper is well written with the ideas clearly explained. The proposed algorithm is well-motivated and backed by strong theoretical guarantees. The experiments show the effectiveness of the proposed approach.

**Weaknesses:**

- The authors have missed an important reference [R1] which considers a nonconvex composite optimization and develops Douglas-Rachford Splitting Algorithms for solving the problem.

- The authors should also discuss [R2] and [R3] which consider a non-convex composite problem but in a decentralized setting and without local updates. Also, in contrast to the duality-based approach taken by the authors, the works [R2] and [R3] propose primal algorithms that directly update the parameters using proximal stochastic gradient descent. The dual approach proposed by the authors is justified by the "curse of primal averaging". A question I have is why the algorithms [R2] and [R3] seem to work even though they are primal algorithms.

- Why are the guarantees presented in the paper independent of the number of clients? The effect of the number of clients should be discussed after the main results. Importantly, does the proposed algorithm achieve linear speed-up with the number of clients in the network?

- In the initial part of the paper the authors refer to the distance-generating function to be strictly convex but later it is assumed to be strongly convex. It is advisable to call it strongly convex from the beginning.

- Define $h_1$, $h_2$ in Definition 3.

- After Definition 3, the authors mention that the previous approaches that add the composite term to the Bregman
divergence may not work for dual extrapolation as certain parts of the analysis break down. Can the authors be more specific about what they mean here?

[R1] Dinh et al., FedDR – Randomized Douglas-Rachford Splitting Algorithms for Nonconvex Federated Composite Optimization, 2021(https://arxiv.org/pdf/2103.03452.pdf)

[R2] Yan et al., Compressed Decentralized Proximal Stochastic Gradient Method for Nonconvex Composite Problems with Heterogeneous Data, 2023 (https://arxiv.org/pdf/2302.14252.pdf)

[R3] Xiao et al., A One-Sample Decentralized Proximal Algorithm for Non-Convex Stochastic Composite Optimization, 2023 (https://arxiv.org/pdf/2302.09766.pdf)

**Questions:**

See the weaknesses section above.

---

> ### Author Response · Authors · 2023-11-15
> **[1/2] Response to Reviewer geTM**
>
> We thank the reviewer for acknowledging our contributions and providing insightful comments. We are more than happy to address these comments and sincerely hope that the reviewer can consider raising the score if the following response helps resolve some concerns.
>
> > The authors have missed an important reference [R1]
>
> We thank the reviewer for bringing this work to our attention and have included it in the updated Appendix B.3. We would note that [R1] solves minimization problems whereas we work on min-max saddle point problems.
>
> > The authors should also discuss [R2] and [R3] which consider a non-convex composite problem but in a decentralized setting and without local updates.
>
> We again thank the reviewer for bringing these results to our attention and have included them in the updated Appendix B.4. We would again note that [R2] and [R3] deal with minimization problems whereas we work on min-max saddle point problems. And, as the reviewer pointed out, [R2] and [R3] are for decentralized optimization without local updates, whereas we focus on the server-client type of distributed optimization with local updates.
>
> > the works [R2] and [R3] propose primal algorithms that directly update the parameters using proximal stochastic gradient descent. ... A question I have is why the algorithms [R2] and [R3] seem to work even though they are primal algorithms.
>
> [R2] and [R3] measure the convergence only in terms of function value, not the structure of the solution. E.g. in Section 5 of [R2] and [R3], $l\_1$ regularization is considered for inducing sparsity, but no explicit measure of sparsity is provided. If observing only the function value, the solution can be dense but still have a small $\ell\_1$ norm, because it's averaged by the number of machines.
>
>
> * For example, assume the solutions on each machine is $x\_1 = [1, 0, ..., 0], x\_2 = [0, 1, ..., 0], ..., x\_{100} = [0, 0, ..., 1]$ for 100 machines, the final averaged solution is $\bar{x} = [0.01, 0.01, ..., 0.01]$. In terms of the function value, the contribution from the regularization, $\Vert\bar{x}\Vert\_1 = 1$, is the same as the solution on each machine, but $\bar{x}$ is apparently not a sparse solution.
>
> In terms of function value convergence, distributed primal-averaging algorithms may indeed converge. This is also observed for the FedMiP algorithm we proposed as a baseline. It shares a similar convergence rate as FeDualEx and is observed to converge in terms of the duality gap as shown in Figure 4. However, as Figure 4 also shows, the sparsity / low rankness of the FedMiP solution differs greatly from that of FeDualEx.

---

> ### Author Response · Authors · 2023-11-15
> **[2/2] Response to Reviewer geTM**
>
> > Why are the guarantees presented in the paper independent of the number of clients? The effect of the number of clients should be discussed after the main results.
>
> As shown in Theorem 1, the final rate is also dependent on $M$, i.e., the number of clients. In particular, the noise term $\frac{5^\frac{1}{2}\sigma B^\frac{1}{2}}{M^\frac{1}{2}R^\frac{1}{2}K^\frac{1}{2}}$ decays with $M$. Since we mainly focus on communication complexity in the context of distributed learning with local updates, the previous rate in Table 1 took the dominating term with respect to the communication rounds $R$, assuming $M$ is large enough, as previously noted in the caption. We have updated Table 1, as well as the discussion after Theorem 1, to clarify this dependence on $M$.
>
> > Importantly, does the proposed algorithm achieve linear speed-up with the number of clients in the network?
>
> In the case of distributed composite convex optimization, we achieve linear speed-up with respect to $M$, because when we assume $R$ to be large, the $O(\frac{1}{\sqrt{MKR}})$ term takes dominance over the $O(\frac{1}{R^\frac{2}{3}})$ term, leading to a linear speed up.
>
> As for distributed composite saddle point optimization, linear speed-up is so far not guaranteed, as there is instead a $O(\frac{1}{R^\frac{1}{2}})$ dependence, which makes $O(\frac{1}{\sqrt{MKR}})$ not able to dominate.
>
> It is important to note, however, we provide the first rate for composite saddle point optimization in the distributed setting. Thus, it remains to be seen whether in this setting linear speedup w.r.t. the number of clients is achievable, and we would consider this study for future work.
>
> > In the initial part of the paper the authors refer to the distance-generating function to be strictly convex but later it is assumed to be strongly convex. It is advisable to call it strongly convex from the beginning.
>
> While our intent was to provide a general definition of Bregman divergence (which only requires strict convexity), we agree it would be clearer in this context to assume strong convexity from the beginning, and so we have updated this accordingly.
>
> > Define $h\_1$, $h\_2$ in Definition 3.
>
> $h\_1$ and $h\_2$ are just any distance-generating functions chosen respectively for $x$ and $y$ in the saddle point function. We have included this in the updated Definition 3.
>
> > After Definition 3, the authors mention that the previous approaches that add the composite term to the Bregman divergence may not work for dual extrapolation as certain parts of the analysis break down. Can the authors be more specific about what they mean here?
>
> Though the exact reason is somewhat technically involved, we provide here a more detailed summary. In short, by ``certain parts of the analysis break down'',  we mean that by using previous definitions other than our Definition 4 for composite dual extrapolation, there will be extra composite terms in the analysis that neither cancel out nor form telescoping terms, but only accumulate and hinder the $O(1/T)$ result (even for the simplest sequential deterministic case as in Section 5 Theorem 4) expected for composite dual extrapolation. If the reviewer would like a more precise mathematical explanation, we would be happy to provide the derivations in full detail.

---

> > ### Comment · Reviewer_geTM · 2023-11-22
> > **Thank you for the response**
> >
> > I thank the authors for the responses. Overall, I am satisfied with the responses.

---

### Official Review · Reviewer_rzX1 · 2023-11-01

**Soundness:** 2 fair
**Presentation:** 2 fair
**Contribution:** 3 good
**Rating:** 5
**Confidence:** 4

**Summary:**

In this paper, the authors study composite saddle-point problems in a Federated learning setup. They propose distributed gradient methods with local updates, which they call Federated Dual Extrapolation. They provide convergence analysis and communication complexity in the homogeneous case.

**Strengths:**

The authors propose a new method, for which they provide convergence analysis. This method has their own interest.

**Weaknesses:**

Table 1 presents the previous and current results strangely:
1) First of all, to compare the obtained complexity for the proposed method with the previous result in a strongly-convex concave case, it should be used standard regularization trick.
2) From my point of you, when complexity contains several terms, each of them should be added.

About Table 2, The authors claim that "The sequential version of FeDualEx leads to the stochastic dual extrapolation for CO and yields, to our knowledge, the first convergence rate for the stochastic optimization of composite SPP in non-Euclidean settings ." It is not true, there is a wide field related to operator splitting in deterministic and stochastic cases. Look at this paper please https://epubs.siam.org/doi/epdf/10.1137/20M1381678.

Also, compared to the previous works, the authors use bounded stochastic gradient assumption and homogeneity of data. In many federated learning papers, those assumptions are avoided. Despite that the authors write "Assumption e is a standard assumption", it would be better to provide analysis without it to have more generality.

In Theorem 1, and Theorem 2, the final result contains mistakes in complexity, because some of them were done in the proof.
The first mistake is made in theorem 3 and repeats in the main theorem. Please look at the last inequality on page 40:
To make $3\eta^2\beta^2 -1 \leq 0$, the stepsize should be chosen in the following way: $\eta \leq \frac{1}{\sqrt{3}\beta}$. This will change the complexity of the methods. The same was done in the proof of Theorems 1, and 2. Please see Lemma 3, 17.

The second mistake is made in the proof of Lemma 13, in the last two inequalities, where should be $\dots\sqrt{2V^l_z(\cdot)} \leq \dots \sqrt{B}$. This thing also will change the final complexity.

The appendix is hard to read in terms of the order of Lemmas. I think it would be better if the numeration of Lemmas had a strict order (for example, after Lemma 5 lemma 6 follows.)

Other things dealing with weaknesses, please, see in questions.

**Questions:**

1. In section 4, it is unclear how you define $\ell_{r,k}$. Could you add an exact expression for it from the appendix to the main part?

Small typos:
1. on the bottom of page 5 in the second argmin the bracket is missed.
2. In definition 4, $t\eta$ is missed in the formula for subgradient.

**Details Of Ethics Concerns:**

-

---

> ### Author Response · Authors · 2023-11-15
> **[1/2] Response to Reviewer rzX1 on main results**
>
> We thank the reviewer for the comments. In the meantime,  we sincerely hope that the reviewer can consider raising the score if the following response helps highlight our contributions and clarify some of the technical details.
>
> ---
>
> We begin by addressing the reviewer's concerns regarding our main results and assumptions.
>
> > Table 1 presents the previous and current results strangely ... 1. to compare the obtained complexity for the proposed method with the previous result in a strongly-convex concave case ... 2. when complexity contains several terms, each of them should be added ...
>
> We would kindly bring to the reviewer's attention that we are presenting a result of a new problem, i.e., distributed **composite** saddle point optimization, instead of comparing with or outperforming existing convergence rates. Table 1 is to show that previous methods either focus on composite convex optimization (instead of saddle point optimization) or non-composite problems in the Euclidean setting, as we highlighted in the second column. As we have noted in the caption, they are listed only for completeness, not for comparison, because they are simply not applicable to composite SPPs (i.e. SPPs with non-smooth regularization).
>
> As for the complete convergence rates, they were omitted simply due to the space limit of the page. We have updated Table 1 to include the full rates.
>
> We sincerely hope that the flaws in the presentation do not overshadow the core contribution of this paper, that is, **we present the first convergence rate for composite SPP with non-smooth regularization under the distributed paradigm.**
>
> > About Table 2 ... It is not true, there is a wide field related to operator splitting in deterministic and stochastic cases. Look at this paper please https://epubs.siam.org/doi/epdf/10.1137/20M1381678.
>
> We would again kindly bring to the reviewer's attention that one of the core components of this paper is **composite optimization**, that is, we study SPPs with composite non-smooth regularization. Based on our understanding of the paper in the link provided, their result does not handle the composite setting. As a result, we would greatly appreciate it if the reviewer could specify which aspects of the paper they believe make our claim untrue.
>
> As for their (smooth non-composite) setting, we have cited earlier work in the bottom right block of Table 2, i.e., stochastic Mirror Prox by Juditsky et al., 2011, though we have also included a citation for the work mentioned by the reviewer in Appendix B.2.
>
> We further emphasize that it is nontrivial to take composite terms into consideration for saddle point problems. Even in the deterministic case, for example, the work composite Mirror Prox (CoMP) by He et al. (2015) shows the degree of technical involvement for extending the original Mirror Prox of Nemirovski (2004) to the composite setting.
>
>
>
> > In many federated learning papers, those assumptions are avoided. Despite that the authors write "Assumption e is a standard assumption", it would be better to provide analysis without it to have more generality.
>
> We wish to note that we are not saying the bounded gradient assumption is a standard assumption in general federated learning, but a standard assumption in distributed composite optimization, even in the recent Federated Composite Optimization by Yuan et al., 2021. (e.g. in their Theorem 4.2 and Assumption 3 in their Appendix D). While we agree with the reviewer that the result would be more general without these assumptions, we wish to retain our focus on dealing with composite terms and saddle point optimization, and hope the reviewer can also acknowledge the technical difficulties we discussed in the ``Remark On Heterogeneity'' at the end of Section 4.2 on page 7.
>
> ---
>
> References:
>
> He et al. "Mirror prox algorithm for multi-term composite minimization and semi-separable problems." Computational Optimization and Applications, 2015.
>
> Nemirovski, Arkadi. "Prox-method with rate of convergence O (1/t) for variational inequalities with Lipschitz continuous monotone operators and smooth convex-concave saddle point problems." SIAM Journal on Optimization, 2004.
>
> Yuan et al. "Federated composite optimization." International Conference on Machine Learning, 2021.

---

> > ### Comment · Reviewer_rzX1 · 2023-12-05
> >
> > Thank you for your response! I will rise my score to 5
> >
> > There are several issues, which are still unclear for me:
> >
> > 1. About Table 1, I have looked at the updates version. If you want to compare your results in convex-concave case with previous results in strongly convex- strongly concave case, the results of other work will have to be changed for the first case via regularization technique. Let me explain it  in details: we rewrite the problem (use the second norm because in those works the authors study Euclidian case):
> > $$\min_{x \in \mathcal{X}}\max_{y \in \mathcal{Y}} f(x, y) + \frac{\varepsilon}{8 B}||x - x^0||^2 -\frac{\varepsilon}{8B}||y - y^0||^2,$$
> > Now we have that the objective function is $\frac{{\varepsilon}{4B}$-strongly convex-strongly concave.
> > For more details see Section 3.1 from https://arxiv.org/pdf/2010.13112.pdf.
> >
> > Also, I can recommend you to provide analysis of your proposed methods in strongly convex-strongly concave case and compare obtained complexity with previous methods.
> >
> > 2. I do not understand fully the main challenge to add composite term. For example, in optimization it is easily doable via non-expansiveness of prox operator. Also for variational inequalities it is well-known how to add prox operator to the method and provide analysis: see Forward-Backward method,  Tseng's Forward-Backward method, Forward-Reflected-Backward method.
> >
> >
> > 3. About bounded gradient: it was standard assumption when community tried to obtain the first results, but now in many papers, they do not use it. For example, now for local methods they also do not use it, especially, look at  Section 2.2 from this work http://proceedings.mlr.press/v108/bayoumi20a/bayoumi20a.pdf. In this paper the authors explain the reason why bounded gradient assumption should be avoided.

---

> ### Author Response · Authors · 2023-11-15
> **[2/2] Response to Reviewer rzX1 on technical details**
>
> We would sincerely thank the reviewer for their meticulous review and would like to clarify the typos and technical details.
>
> > In Theorem 1, and Theorem 2, the final result contains mistakes in complexity ... The second mistake is made in the proof of Lemma 13 ...
>
>
> These are indeed typos, but only the constants were affected, and **the rate is in fact better after fixing these typos**. Because we would note that the first one only affects the power on $\beta$. After fixing the typo, Theorem 3 becomes
> $$\mathbb{E}[Gap] \leq \frac{{\color{red}\sqrt{3}\beta} B}{T} + \frac{3^\frac{1}{2}}{T^\frac{1}{2}}.$$
> Only the part in red is changed from $3 \beta^2$ to $\sqrt{3} \beta$. Similarly, for Theorem 1 and 2, choosing $\eta^c \leq \frac{1}{\sqrt{5}\beta}$ instead of $\frac{1}{5\beta^2}$ only changes the first term in Theorem 1 and 2 from $\frac{5\beta^2 B}{RK}$ to $\frac{\sqrt{5}\beta B}{RK}$.
>
> The second one only affects the power on $B$ in the last term of Theorem 1. Letting $\eta^c \leq \frac{B^\frac{1}{4}}{2^\frac{3}{4}\beta^\frac{1}{2}G^\frac{1}{2}K R^\frac{1}{2}}$ instead makes the last term of Theorem 1 become $\frac{2^\frac{3}{4}\beta^\frac{1}{2}G^\frac{1}{2}{\color{red}{B}^\frac{3}{4}}}{R^\frac{1}{2}}$. Only the part in red is changed from $B$ to ${B}^\frac{3}{4}$.
>
> **We have fixed these typos throughout the paper in the updated pdf.**
>
> > The appendix is hard to read in terms of the order of Lemmas.
>
> We apologize for the confusion caused by the ordering of the lemmas. The intention was to number the main lemmas used for Theorem 1 as Lemma 1,2,3 (i.e., the Lemma 11 and 12 on pages 30 and 31 should in fact be Lemma 1 and 2, and this is now fixed). We will certainly re-work the numbering but leave it unchanged for the moment for the convenience of the on-going discussion period.
>
> > Questions: In section 4, it is unclear how you define $\ell\_{r,k}$
>
> $\ell\_{r,k}$ is formally given on the 6th line of the ``Projection Reformulation'' paragraph under Section 4.2. It simply breaks $t$ into the number of communication rounds $r$ and the number of local updates $k$. Since $\ell\_t(z) = l(z) + t \eta\psi(z)$, $\ell\_{r,k}(z) = l(z) + (rK\eta^s + k)\eta^c\psi(z)$ for $t = rK + k$, in which $r$ is the round of communications till now, $K$ is the number of local updates between each round, and $k$ is the number of local updates since last communication.

---

> > ### Comment · Reviewer_rzX1 · 2023-11-22
> >
> > Thank you for your response to addressing issues! Now complexity looks like as expected.

---

> > > ### Author Response · Authors · 2023-11-27
> > >
> > > We thank the reviewer for their kind response and acknowledgement! Based on this response, we believe we have now addressed the reviewer's comments, though we are happy to address any further concerns. Otherwise, we would kindly ask the reviewer to consider raising their score.

---

> ### Author Response · Authors · 2023-11-21
> **Addressing further concerns**
>
> Dear Reviewer rzX1,
>
> We appreciate your helpful comments, and we believe we have addressed your concerns in our rebuttal. As the deadline for the discussion period is approaching, please let us know if there are any further concerns to address, and thank you once again for your time and effort in this matter.
>
> Sincerely,
> Authors

---

### Official Review · Reviewer_jzbD · 2023-11-26

**Soundness:** 3 good
**Presentation:** 3 good
**Contribution:** 3 good
**Rating:** 6
**Confidence:** 3

**Summary:**

In this paper, the authors propose an Algorithm FeDualEx for solving composite saddle point problems under distributed settings. The proposed algorithm is inspired from the dual extrapolation algorithm while using a proximal operator which they define using the generalized Bregman divergence defined for saddle functions. They analyze this algorithm under homogeneous settings and derive its convergence rate for the duality gap. They also study the special cases when the number of clients equals 1, where the convergence rate of FeDualEx matches the existing rates known in the literature. The study also demonstrates that solving using the dual extrapolation has advantages of learning better sparse solutions than solving the primal.

**Strengths:**

The paper studies federated learning of composite saddle point problems, for which there does not seem to be much existing work. The proposed convergence rates.

(Novelty) The paper proposes a new bregman divergence for saddle functions and its associated proximal operator, which are used in the dual extrapolation steps.

(Clarity) The main results are presented well and contrasted to the related ones. The experimental results illustrate the benefit of solving using the federated dual extrapolation over methods such as Federated Mirror Prox. The comparison to the sequential algorithms also help to position the contributions in relation to the existing work.

**Weaknesses:**

The algorithm is similar that of Federated Dual Averaging (Yuan et.al.) while incorporating the dual extrapolation strategy over the newly defined Bregmen divergence and the proximal operators. The challenges associated with adapting the above strategy over FeDualAvg doesn't seem to be conveyed well in the paper.

From the motivations perspective, some examples of practical setups which required distributed learning of saddle point formulations would be useful in appreciating the contributions better.

**Questions:**

Questions / Comments
Is it possible to have a similar algorithm only using the generalized bregman divergence on x ?
Some discussion on the relation between the variables z=(x,y) and ς would help since the former already includes a primal and dual pair.

To improve the clarity, one may include convexity assumptions of the functions involved while the main problem is defined in (1).

---

> ### Author Response · Authors · 2023-11-27
> **Response to Reviewer jzbD**
>
> We thank the reviewer for their insightful comments, as well as for kindly acknowledging our contributions. We are more than happy to address these comments and hope that the following response helps resolve some concerns.
>
> > The algorithm is similar that of Federated Dual Averaging (Yuan et.al.) while incorporating the dual extrapolation strategy over the newly defined Bregmen divergence and the proximal operators. The challenges associated with adapting the above strategy over FeDualAvg doesn't seem to be conveyed well in the paper.
>
> While sharing some of the motivations behind FedDualAvg, composite saddle point optimization is more general than convex optimization (since it involves a min-max objective), and thus algorithms suited to this more general problem (e.g., dual extrapolation) involve more steps than their counterparts for convex optimization (e.g., dual averaging). As such, dealing with these additional considerations (along with the previous technical challenges in FedDualAvg, e.g., handling the distributed, composite, non-Euclidean setting) turns out to be rather non-trivial, as the techniques that might naturally be suited for one of these settings would fail for another. We have briefly addressed this point in Section 4.1 ("The smooth analysis of dual extrapolation is already non-trivial ... and no attempts were previously made for generalizing dual extrapolation to the composite optimization realm"), though we will certainly emphasize these difficulties further in the revised version.
>
> As the reviewer kindly acknowledges, we proposed "the newly defined Bregman divergence and the proximal operators" to overcome the difficulties in the analysis that arise when using previous definitions. Though the exact technical challenge is somewhat involved, we provide here a more detailed summary. In short, using previous definitions other than our Definition 3 and 4 for composite dual extrapolation, there will be extra composite terms in the analysis that neither cancel out nor form telescoping terms, but only accumulate and thus hinder the result (even for the simplest sequential deterministic case as in Section 5 Theorem 4) expected for composite dual extrapolation. If the reviewer would like a more precise mathematical explanation, we would be happy to provide the derivations in full detail.
>
> > From the motivations perspective, some examples of practical setups which required distributed learning of saddle point formulations would be useful in appreciating the contributions better.
>
> We have included in our experiments (as detailed in the last two paragraphs of Section 6) a real-world example of distributed adversarial training (which is formulated as a saddle point problem, as given in Appendix A.3). Other relevant SPPs include GANs, matrix games, and multi-agent reinforcement learning, as mentioned in the Introduction. These examples highlight the importance of developing more efficient, distributed methods for saddle point problems, particularly when faced with increasing amounts of (privacy sensitive) data, which might indeed require distributed learning to avoid the centralization of local data (or collecting personal data). We will include such motivating discussion in the revised Introduction.
>
> ---
>
> We thank the reviewer for the questions and comments on the presentation. We address them here and will revise our paper accordingly, as it seems that we're currently unable to make any edits to the pdf.
>
> > Is it possible to have a similar algorithm only using the generalized bregman divergence on x? Some discussion on the relation between the variables z=(x,y) and $\varsigma$ would help since the former already includes a primal and dual pair.
>
> It's possible to have the algorithm for $x$ only, in which case it reduces to an algorithm for composite convex optimization (instead of saddle point optimization). In fact, we have already included the convergence rate for this case ($x$ only) in our Theorem 2 with detailed derivation in Appendix F. We will include the algorithm in the revised version.
>
> We will make it clear in the revised version that $\varsigma = (\mu, \nu)$ is the dual image of $z = (x, y)$, in which $x, y$ correspond to the variables for minimization and maximization in the SPP respectively, and $\mu, \nu$ are their counterparts in the dual space. The relation between $z$ and $\varsigma$ is briefly mentioned in Definitions 3, 4, and Definition 2 for $x$ and $\mu$. The general notation of using English letters for primal-space variables and Greek letters for dual-space variables is introduced at the beginning of Section 3.
>
> > To improve the clarity, one may include convexity assumptions of the functions involved while the main problem is defined in (1).
>
> We will include the convex-concave assumption in Definition 1 in the revised version.

---

### Author Response · Authors · 2023-11-19

We thank the reviewers for their insightful discussions and hope that our latest responses have addressed their concerns. If the reviewers have any further questions or comments, we would be happy to provide additional clarification.

---

### Author Response · Authors · 2023-11-30

We thank all reviewers for their insightful discussions and hope that our latest responses have addressed their concerns. As the extended deadline for the discussion period is approaching, please let us know if there are any further concerns to address, and thank you once again for your time and effort.
Sincerely, Authors

---

### Meta-Review · Area_Chair_K3ok · 2023-12-11

**Metareview:**

The paper contributes FeDualEx, a new algorithm for solving composite saddle point problems in the distributed setting. Both convergence rates and communication complexity of the algorithm is discussed.
While there is a substantial body of work on saddle point problems, the work on composite saddle point
problems is limited.
The makes progress over FedDualAvg(the existing state of the art), however a detailed discussion on the points where FeDualEx improves over the state of the art is missing. The final paper(if it is accepted) need to reflect this discussion.

**Justification For Why Not Higher Score:**

It is a very niche topic(composite saddle point optimisation in distributed setting) and hence maybe of very limited interest.
More discussion on how the paper improves over FedDualAvg would have been helpful.

**Justification For Why Not Lower Score:**

The authors have satisfactorily answered the key queries during rebuttal.

---

### Decision · Program_Chairs · 2024-01-16

Accept (poster)